# A global long-term (1981-2000) land surface temperature product for NOAA AVHRR

Jin Ma[1,2], Ji Zhou[1] [*], Frank-Michael Göttsche[2], Shunlin Liang[3], Shaofei Wang[1], Mingsong Li[1]

[1]School of Resources and Environment, Center for Information Geoscience, University of Electronic Science and Technology of China, Chengdu 611731, China

[2]Institute of Meteorology and Climate Research, Karlsruhe Institute of Technology, Karlsruhe 76344, Germany

[3]Department of Geographical Sciences, University of Maryland, College Park 20742, USA

*Correspondence to*: J. Zhou (jzhou233@uestc.edu.cn)

**Abstract**, Land Surface Temperature (LST) plays an important role in the research of climate change and various land surface processes. Before 2000, global LST products with relatively high temporal and spatial resolutions are scarce, despite of a variety of operational satellite LST products. In this study, a global $0.05\degree \times 0.05\degree$ historical LST product is generated from NOAA AVHRR data (1981-2000), which includes three data layers: (1) instantaneous LST, a product generated by integrating several Split-Window Algorithms with a Random Forest (RF-SWA); (2) orbital drift corrected (ODC) LST, a drift corrected version of RF-SWA LST; (3) monthly averages of ODC LST. For an assumed maximum uncertainty in emissivity and column water vapour content of 0.04 and 1.0 g/cm$^2$, respectively and evaluated against the simulation data set, the RF-SWA method has a Mean Bias Error (MBE) of less than 0.10 K and a Standard Deviation (STD) of 1.10 K. To compensate the influence of orbital drift on LST, the retrieved RF-SWA LST was normalized with an improved ODC method. The RF-SWA LST were validated with *in-situ* LST from Surface Radiation Budget (SURFRAD) sites and water temperatures obtained from the National Data Buoy Center (NDBC). Against the *in-situ* LST, the RF-SWA LST has a MBE 0.03 K with a range of -1.59 K – 2.71 K and STD is 1.18 K with a range of 0.84 K – 2.76 K. Since water temperature only changes slowly, the validation of ODC LST was limited to SURFRAD sites, for which the MBE is 0.54 K with a range of -1.05 K to 3.01 K and STD is 3.57 K with a range of 2.34 K to 3.69 K, indicating a good product accuracy. As global historical datasets, the new AVHRR LST products are useful for filling the gaps in long-term LST data. Furthermore, the new LST products can be used as input to related land surface models and environmental applications. Furthermore, in support of the scientific research community, the
datasets are freely available at https://doi.org/10.5281/zenodo.3934354 for RF-SWA LST (Ma et al., 2020a); https://doi.org/10.5281/zenodo.3936627 for ODC LST (Ma et al., 2020c); https://doi.org/10.5281/zenodo.3936641 for monthly averaged LST (Ma et al., 2020b).

## 1. Introduction

Land surface temperature (LST) is an important parameter for energy exchange between Earth's surface and the atmosphere

and, thus, an important indicator for global climate change. Therefore, LST has been widely used in research and applications of land surface processes and models, e.g. climate and meteorology, hydrology, and disaster monitoring (Anderson et al., 2011; Jin and Dickinson, 2002; Van Der Werf et al., 2017). Compared to traditional ground observations, retrieving LST from remote sensing is an effective way of taking advantage of the spatio-temporal coverage offered by satellites. Since the 1970s, the accurate retrieval of LST from satellite has been an active area of research in quantitative remote sensing. The main sources

for retrieving LST from satellite data are thermal-infrared (TIR) remote sensing and passive microwave (MW) remote sensing (Holmes et al., 2009; Li et al., 2013a), which both are effective means for obtaining the radiance emitted by Earth's surface. Although MW remote sensing is less affected by cloud and fog, it is limited by its coarser spatial resolution and its higher thermal sampling depth compared to TIR remote sensing, which results in a lower retrieval accuracy (Zhou et al., 2017). Therefore, retrieving LST from TIR remote sensing is still the dominant approach, since it offers a better physical definition

and higher retrieval accuracy. LST retrieval from satellite TIR remote sensing is based on the simplification of the radiative transfer model. A variety of algorithms have been proposed for retrieving LST from TIR data, e.g. Split-Windows Algorithms (SWA), Mono-window Algorithms/Single Channel Algorithms and Temperature-Emissivity separation algorithms (TES) (Gillespie et al., 1998; Li et al., 2013a; Wan and Dozier, 1996). Selecting a suitable algorithm for retrieving LST depends on the sensor's number of TIR channels and their spectral specifications.

The SWA is a good choice for retrieving LST from sensors with two or more TIR channels centred at 11 μm and 12 μm, e.g. Terra/Aqua MODIS and Sentinel-3/SLSTR. Based on the idea that the atmospheric absorption in the thermal band can be related to the brightness temperature (BT) difference between two adjacent channels, McMillin (1975) initially proposed the SWA for retrieving sea surface temperature (SST) from NOAA/AVHRR. SWAs for retrieving SST from various sensors were



developed, which were based on different assumptions (Llewellyn-Jones et al., 1984; Niclòs et al., 2007). Inspired by the

success of the SST algorithm, the first SWA for retrieving LST was proposed by Price (1984). However, in contrast to nearly

homogeneous and isothermal water bodies, LST is affected by multiple additional factors, e.g. land cover type (LCT), material

dependent emissivity, terrain, and viewing geometry. Therefore, one or more terms were added to the basic SWA to describe

these effects, e.g. land surface emissivity (Wan, 2014), vegetation cover fraction (Prata, 2002), view zenith angle (Becker and

Li, 1990a), and water vapour (Sobrino et al., 1991). Nevertheless, there are still limitations in LST retrieval with SWAs (Li et

al., 2013a), e.g. the requirement for a priori knowledge of emissivity and a dependence of LST retrieval accuracy on SW

coefficients, which in turn depend on observation and atmospheric conditions. Furthermore, due to the variation of land surface

and atmospheric conditions, no single SWA performs the best under all conditions.

      Currently, several LST products retrieved from satellite TIR remote sensing are available. Global LST products for

Terra/Aqua MODIS are available since 2000, e.g. M*D11/M*D21 (Hulley and Hook, 2011; Wan, 2014). Similarly, a JPSS-

VIIRS LST product is available since 2012 (Guillevic et al., 2014) and China FengYun-VIRR LST is available since 2009

(Dong et al., 2012). The aforementioned sensors observe Earth's surface twice per day with a spatial resolution of ~1 km at

nadir. For the user's convenience, some LST products are processed into different temporal and spatial resolutions, e.g. daily

/ monthly and 1 km×1 km / 0.05 °×0.05 °. The operational LST product retrieved from the (A)ATSR series between 1995 and

2012 is a typical SWA LST product (Prata, 2002). AATSR's nadir spatial resolution onboard ENVISAT was approximately 1

65    km and its temporal resolution 3 days. Since 2016, its successor, SLSTR onboard Sentinel-3 A and B, provides daily temporal

resolution and a consistent spatial resolution (Ghent et al., 2017). Global LSTs retrieved from satellite TIR also include Landsat

LST (Parastatidis et al., 2017) and ASTER LST (Hulley and Hook, 2011), which have significantly higher spatial resolutions

(e.g. about 100 m), but considerably lower temporal resolutions (e.g. every 16 days). LST products from geostationary satellites

are generated at lower spatial resolution (3-5 km) but considerably higher temporal resolution (10 – 60 min), e.g. GOES-ABI

LST for the Americas and Africa (Yu et al., 2009), MSG-MVIRI/SEVIRI LST for Europe, Africa, and the Atlantic Ocean

(Duguay-Tetzlaff et al., 2015; Trigo et al., 2008), FY- SVISSR/AGRI LST and Himawari-AHI LST for the Asian-Pacific

region (Choi and Suh, 2018; Jiang and Liu, 2014). Dech et al. (1998) and Pinheiro et al. (2006) provide African and European

LST for NOAA-14 AVHRR, Zhou et al. (2019a) provide an all-weather LST product retrieved from combined TIR and



MW/Reanalysis data over the Tibetan plateau from 2003 to 2018. There are also a few LST products from MW (e.g. for SSM/I
and AMSR-E) (Aires et al., 2001; Jiménez et al., 2017), and LST for land surface models (e.g. ECMWF and GLDAS) (Fang
et al., 2009; Viterbo and Beljaars, 1995); however, these temperature products have lower spatial resolutions and slightly
different meanings than TIR LST. In summary, from 1991 onwards, many global and regional satellite LST products are
available, but higher spatio-temporal resolutions (e.g. 1 km - daily) are only available after 2000. At the same time, many
climate applications urgently need higher spatial-temporal resolution LST products for the time before 2000. It has been
reported that 1983-2012 were the warmest 30 years for nearly 1400 years (IPCC, 2014). The warm climate change trend has
also caused changes in many land surface processes, e.g. most glaciers on the Tibetan Plateau are in retreat (Yao et al., 2012)
and cover progressively smaller areas. The LST around glaciers is a highly useful indicator of this phenomenon and allows
predicting trends in glacier status (Steiner et al., 2008). Similar demands for LST data also exist in global drought monitoring
(Sánchez et al., 2018), studies of species distribution (Lembrechts et al., 2019), and land surface modelling (Bechtel, 2012;
Ghent et al., 2017; Reichle et al., 2010). Therefore, it is meaningful to extend the global LST time series with a relatively high
spatio-temporal resolution (i.e. 5 km - daily) to the historical NOAA AVHRR data before 2000.

A major factor limiting applications of AVHRR LST is orbital drift, which over the lifespan of the NOAA satellites leads
to shifts to later overpass times and, therefore, affects temporal comparability. Two main approaches were developed to remove
the effect of orbital drift. On the one hand, based on the regular diurnal temperature variation typically observed under clear-
sky, several researchers corrected orbital drift by fitting a diurnal temperature cycle (DTC) model to reanalysis or geostationary
datasets (Jin and Treadon, 2003; Parton and Logan, 1981) and then normalizing LST to a given time. On the other hand, a
relationship between LST anomaly and solar zenith angle was used for correcting LST to a given solar zenith angle (Gleason
et al., 2002; Gutman, 1999; Julien and Sobrino, 2012). Various applications made use of the two types of orbital drift correction
methods for AVHRR LST, but a general method for global application is still missing, i.e. the former method suffers from the
low spatial resolution of its input datasets, while latter leads to inconsistent times. Liu et al. (2019a) proposed another method
for correcting AHVRR LST orbital drift, which fits a DTC model to component temperatures of neighbourhood pixels and
was reported to achieve good accuracy. This approach is used here to generate a time-consistent AVHRR LST product.

The objective of this study is to develop a long-term global LST product (1981-2000) from historical NOAA AVHRR

datasets. Section 2 describes the simulation datasets used for developing consistent SWAs, the input datasets for LST product

generation, and the *in-situ* datasets for validating the retrieved LST. In section 3, a practical approach for generating a single

optimized LST product is proposed, which integrate several well-established SWAs through the Random Forest, which is

termed RF-SWA. Finally, the retrieved RF-SWA LST is normalized with an improved orbital drift correction method.

Furthermore, emissivity estimation for bare soil is improved by using ASTER Global Emissivity Dataset (GED) and yields

more accurate estimates of land surface emissivity in section 3.3. Section 4 describes the results and provides implementation

details of the LST retrieval method, LST validation, and give an example of the LST product. Data availability shows in section

5. Conclusions and outlooks are provided in section 6.

## 2. Datasets

### 2.1 Satellite remote sensing datasets

#### 2.1.1 AVHRR datasets

The advanced very-high-resolution radiometer (AVHRR) is a sensor onboard NOAA polar-orbiting satellite series. The orbital

period is 101.4 minutes and designed over-pass time at the equator is between 13:30 and 14:30 (solar time) depending on the

satellite. The second AVHRR version (AVHRR/2) has five spectral channels, including a visible band (0.55-0.68 μm), a near-

infrared band (0.75-1.1 μm), a middle-infrared band (3.55-3.93 μm), and two thermal bands (10.5-11.3 μm and 11.5-12.5μm).

Figure 1 shows the spectral responses of the two AVHRR thermal channels of NOAA-07/09/11/14. Nadir spatial resolution of

the TIR channels is 1.1 km×1.1 km, and scan angles range between -55 ° and 55 °. AVHRR covers the Earth's surface twice

daily and has been widely used to generate various local or global land/sea surface parameters, e.g. the normalized difference

vegetation index (NDVI) and SST. In this study, the AVHRR datasets from Long-Term Datasets Records (Pedelty et al., 2007)

(LTDR, https://ltdr.modaps.eosdis.nasa.gov/) are used, including AVH02C1 and AVH13C1 (Table 1). Those two datasets

include the top-of-atmosphere BT of the TIR channels, NDVI, view zenith angle (VZA), view time, and quality control (QC)

flags, which provide a reference for distinguishing pure and cloudy pixels.





### 2.1.2 ASTER Global Emissivity Dataset (GED)

ASTER onboard the Terra satellite and has five TIR channels (Fig. 1). The ASTER GED used in this study was generated from clear sky ASTER TIR data between 2000 and 2008 with the TES algorithm and the water vapour scaling atmosphere correction method (Hulley et al., 2015). The products are output at 3″ (~100 m) and 30″ (~1 km) spatial resolution on 1°×1° tiles. Channel

temporal mean emissivity, LST, and NDVI, as well as their standard deviation, global DEM, and land-sea mask are part of the GED. In this study, the ASTER GEDv3 with a 1-km spatial resolution was used to determine the global background emissivity of bare land.

### 2.2 Atmospheric profiles and forward simulation datasets

Global forward simulation datasets with good representativeness are necessary for developing and evaluating LST retrieval

algorithms. This requires a reliable atmospheric profile dataset as input. In this study, the well-established SeeBor V5.0 (Borbas et al., 2005) and TIGR2000 V1.2 (Chedin et al., 1985) atmospheric profiles were used to construct the forward simulation datasets. Zhou et al. (2019b) derived a global atmospheric profile dataset (GAPD) by screening the SeeBor V5.0 atmospheric profiles and removing cloud-contaminated and redundant profiles. The GAPD dataset has been used for developing the LST retrieval algorithm for NOAA-20/VIIRS and Sentinel-3/SLSTR (Liu et al., 2019b; Yang et al., 2020): it contains 549 global

profiles with a column water vapour content (CWVC) range of $0.014 - 7.939$ g/cm$^2$ and near-surface air temperature (NSAT) range of 224.25 K – 309.05 K. Here, the GAPD was used to generate a training dataset (TRA-G): globally representative observation conditions were simulated by varying the viewing geometry and land surface characteristics over a realistic range for a limited profile dataset Zhou et al. (2019b), i.e. for each profile 10 surface temperatures ($T_s$), 15 view zenith angles (VZA), and 48 land surface emissivities (LSE, $\varepsilon$) were set. Specifically, $T_s$ was set relative to NSAT with the difference ($T_s$-NSAT)

covering the range of -16 K to +20 K at an interval of 4 K; VZA was set to values from 0 ° to 70 ° at an interval of 5 °; emissivity was obtained from Johns Hopkins University (JHU) spectral emissivity library by convolving the emissivity spectra with the spectral response functions of NOAA-07/09/11/14 AVHRR (Fig. 1); the corresponding emissivity ranges are provided in Table 2. For the remaining 4761 SeeBor clear-sky profiles (ATP-S) and 506 TIGR clear-sky profiles (ATP-T), the corresponding simulations were performed and used as evaluation datasets VAL-S and VAL-T, respectively. In contrast to GAPD, for each

profile in ATP-S (ATP-T), we randomly set 10 (10) VZAs between 0 ° and 70 °. The corresponding LSE has been assigned

according to the LCT over which a profile is located (Snyder et al., 1998) and $T_s$ for VAL-S and VAL-T was set to the

corresponding NSAT. Table 3 summarizes the three profile datasets and the corresponding simulation datasets. More details

can be found in Zhou et al. (2019b) and Yang et al. (2020).

## 2.3 Ancillary data used for LST retrieval

Four ancillary datasets were used for LST retrieval: NSAT, CWVC, LCT, and soil type. The MERRA-2 reanalysis dataset

(M2T1NXINT) provides NSAT and CWVC (variables in datasets: T2M and TQV, respectively) with 0.5 °×0.625 ° spatial

resolution and hourly temporal resolution; nearest neighbour sampling was used to match up with AVHRR pixel and over-pass

time. AVHRR LCTs were obtained from the University of Maryland (UMD) dataset (Defries and Hansen, 2010), which

provides 14 LCTs (0:Water; 1:Evergreen Needleleaf Forest; 2:Evergreen Broadleaf Forest; 3:Deciduous Needleleaf Forest;

4:Deciduous Broadleaf Forest; 5:Mixed Forest; 6:Woodland; 7:Wooded Grassland; 8:Closed Shrubland; 9:Open Shrubland;

10:Grassland; 11:Cropland; 12:Bare Ground; 13:Urban and Built). The spatial resolution of the UMD LCT dataset is 1 km ×

1 km. To adapt its resolution of AVHRR, the dominant LCT within each 0.05 ° grid was used as the LCT for AVHRR. The soil

type dataset employed for estimating AVHRR LSE is provided by the United States Department of Agriculture, which is

mainly based on the world soil map of FAO-UNESCO. Its spatial resolution is 2′ (~ 0.03 °) and the soil type of each AVHRR

pixel was also set to the dominant type.

## 2.4 *In-situ* datasets

*In-situ* measurements from the Surface Radiation Budget (SURFRAD) network and the National Data Buoy Center (NDBC)

were used to validate the retrieved AVHRR LST. The details and geographical distribution of the selected *in-situ* sites are

provided in Table 4 and Fig. 2. SURFRAD was established in 1993 and focuses on validating Earth's radiation budget, but

SURFRAD data also have been widely used for validating satellite-retrieved LST products (Liu et al., 2019b; Martin et al.,

2019). Six sites providing *in-situ* data between 1995 and 2000 were selected. At these sites, upwelling and downwelling

longwave radiances are measured with highly accurate Eppley Precision Infrared Radiometers (PIR; wavelength: 4-50 μm) at

an observation interval of 3 minutes. The PIRs were set up ~10 m above the ground, giving them a field of view (FOV)

covering approximately $70{\times}70$ m$^2$ (Guillevic et al., 2014). Historical data from the NDBC

(https://www.ndbc.noaa.gov/historical_data.shtml) provide hourly samples of bulk water temperature, which is measured with

electronic thermistors. Considering the thermal homogeneity of the water surface, buoy temperatures are usually representative

of the satellite pixel scale, even if it covers large areas. To avoid mixed land-water pixels, only buoys at least 20 km from the

coastline were selected.

## 3. Methodology

LST retrieval algorithm from TIR remote sensing, especially with SWAs, is a well established and validated method. However,

no single algorithm performs best under all conditions, even if it generally achieves good accuracy (Yu et al., 2009). This

suggests that a more stable and robust LST retrieval algorithm may be obtained by integrating various individual LST retrieval

algorithms. In this study, the Random Forest (RF) ensemble method (Breiman, 2001) was utilized for integrating multi-LSTs

(mLSTs) obtained with several-SWAs into a global AVHRR LST product. First, widely used candidate SWAs were trained and

evaluated; these SWAs have been studied in previous work (Yang et al., 2020; Zhou et al., 2019b), and are shown in Table 5

for readers' convenience. Second, estimates of land surface emissivity were improved by combining the NDVI threshold

method and ASTER GED. Third, the LSTs from the trained candidate SWAs were integrated with the RF method: thus, the

approach is termed RF-SWA. Then, the instantaneous RF-SWA LST was normalized to 14:30 (solar time) using an improved

orbital drift correction (ODC) method and the RF-SWA LST and ODC LST products were validated against *in-situ* LST. Finally,

for the user's convenience of users, a monthly averaged LST was also generated from the ODC LST.

### 3.1 Refining the Candidate Algorithms

Forward radiative transfer simulations with PMODTRAN (Berk et al., 2005; Huang et al., 2016) were performed on a high-

performance computing platform (2*Intel @Xeon E5-2650 2.00GHz (8Cores), 64GB 1600MHz) for the datasets GAPD, ATP-

S, and ATP-T described in section 2.2; the corresponding simulated datasets were labelled as TRA-G, VAL-S, and VAL-T.

Each forward simulation yields channel-specific top-of-atmosphere radiances and BTs in dependence of NSAT, CWVC, and

VZA. To simulate BTs measured by satellites more realistically, Gaussian-distributed noise with a noise equivalent differential

temperature (NEDT) of 0.12 K was added to the simulated BTs. More details on the simulations are provided in Zhou et al.

(2019b).

Multiple regression was performed on the simulated training datasets, TRA-G, to determine the coefficients of the

candidate SWAs in Table 5. The TRA-G dataset was divided into 480 groups based on NSAT, CWVC, VZA, and $T_s$-NSAT as

follows: (i) the atmospheres were divided into Cold-ATM and Warm-ATM with a NSAT threshold of 280 K; (ii) the data were

divided into CWVC classes with an interval of 0.5 g/cm$^2$. This resulted in 3 subgroups of Cold-ATM and 13 subgroups of

Warm-ATM; (iii) the VZAs were divided into intervals of 5 °; (iv) based on $T_s$-NSAT, the data were divided into two subgroups

e.g. [-16,4] K and [-4, 20] K, approximately representing daytime and nighttime cases, respectively. Based on regression

against these training datasets, look-up tables (LUT) with coefficients for each candidate SWA were established. The candidate

algorithms were then analyzed w.r.t. the standard error of the estimate (SEE) and coefficient of determination ($R^2$) and a

sensitivity analysis was performed for the main input parameters, e.g. LSE and CWVC, to test the stability and accuracy of

the trained SWAs. Being consistent with the uncertainty level in Zhou et al. (2019b), the various uncertainty sources were

grouped into 2 levels: (i) L1: $|\delta\varepsilon_{11}|_{max} \leq 0.02$, $|\delta\varepsilon_{12}|_{max} \leq 0.02$, and $|\delta CWVC|_{max} \leq 1.0$ g/cm$^2$; (ii) L2: $|\delta\varepsilon_{11}|_{max} \leq 0.04$, $|\delta\varepsilon_{12}|_{max} \leq 0.04$,

and $|\delta CWVC|_{max} \leq 1.0$ g/cm$^2$. These uncertainties will be added to $\varepsilon_{11}$, $\varepsilon_{12}$, and CWVC as random noises. Datasets without

added uncertainty were labelled as L0. All trained candidate algorithms were evaluated against the simulation datasets VAL-S

and VAL-T.

**3.2 Multi-LSTs ensemble**

Based on the training and evaluation results (see section 4.1), a multi-LST ensemble method is proposed, which achieves a

more stable retrieval by integrating the most robust and stable SWAs. The method used to integrate the selected SWAs is the

Random Forest (RF) method proposed by Breiman (2001). Compared to detailed analytic expressions for explaining

complicated nonlinear relationships, the RF method has several outstanding characteristics, including the ability to process

large databases with high efficiency, unbiased estimation, and especially minimizing the risk of overfitting (Hutengs and

Vohland, 2016). Therefore, the RF method has been widely used in remote sensing applications, e.g. land cover classification





(Rodriguez-Galiano et al., 2012), land surface parameter downscaling (Zhao et al., 2018), and estimating vegetation cover

parameters (Mutanga et al., 2012).

The RF method utilizes an ensemble of many decision trees. In the implementation of the RF ensemble method, a random

vector $\Theta_k$ is selected from the input training datasets (mLSTs, LST) with the Bootstrap sampling method. Here, $k$ is the number

of samplings; mLSTs are the LSTs retrieved with the individual SWAs, i.e. the predictors; LST, i.e. the target variable is known

from the forward simulations. The sample size of each sampling is two-thirds of the observations; for each sampling, a tree is

grown using the training set and $\Theta_k$, which results in a tree predictor $T(\Theta_k)$. Finally, the LST predicted with the RF is formed

by averaging over the $k$ trees (Eq. 1),

$$g = \frac{1}{k}\sum_{i=1}^{k} T(\Theta_k) \tag{1}$$

Along with the predicted LST, the importance of each variable can be calculated using the residual sum of squares (RSS),

which usually has larger values for more influential mLSTs. Additionally, the simple average (SA) method and Bayesian Model

Averaging (BMA) method are implemented for comparison. To cover the real natural variability as much as possible, datasets

TRA-G (L0, L1, and L2), VAL-S (L0), and VAL-T (L0) are used as training datasets for the LST ensemble model. The

remaining datasets VAL-S and VAL-T at uncertainty levels L1 and L2 are used for evaluating the ensemble model. For the

later generation of global LST products, only mLSTs from the selected SWAs are needed.

**3.3 Estimating LSE**

LSE is a key parameter in retrieving LST from TIR remote sensing data. Depending on the spectral channels of the sensor and

the available temporal sampling, there are various LSE estimation methods, e.g. the NDVI-threshold method (Sobrino et al.,

2008), land cover-based (LC-based) method (Snyder et al., 1998; Wan, 2014), TES method (Gillespie et al., 1998), day-night

method (Becker and Li, 1990b), and Kalman filter method (Li et al., 2013b; Masiello et al., 2015). The NDVI-threshold and

LC-based methods are widely used in retrieving LST (Sobrino et al., 2008; Wan, 2014). However, those methods require that

the emissivity of the land cover or the vegetation and bare (background) soil is known. In the TIR, emissivity spectra of dense

vegetation are relatively similar and, therefore, can be taken from spectral libraries; these spectra can then be convolved with

the sensor's spectral response functions to obtain channel effective emissivities. In contrast, the emissivity of bare soil varies



considerably, mainly due to variations of its components, roughness, water content, and surface structure. Therefore, this study

employs a practical and robust method that combines the ASTER GED and the NDVI-threshold method to determine LSE.

First, the land surface is classified into pure bare soil, mixture of vegetation and bare soil, and pure vegetation. The

emissivity in mixed areas ($\varepsilon_\lambda$) is obtained as the weighted sum of vegetation emissivity ($\varepsilon_{\lambda,v}$) and bare soil emissivity ($\varepsilon_{\lambda,s}$),

where the fraction of vegetation cover ($f_v$) determines the weights (Carlson and Ripley, 1997; Hulley et al., 2015):

$$\varepsilon_\lambda = \varepsilon_{\lambda,v} f_v + \varepsilon_{\lambda,s}(1 - f_v) \tag{2}$$

here, the $f_v$ can be calculated as Eq. (3),

$$f_v = \begin{cases} 0, & NDVI \leq \mathrm{NDVI_{min}} \\ 1 - \frac{\mathrm{NDVI_{max}} - NDVI}{\mathrm{NDVI_{max}} - \mathrm{NDVI_{min}}}, & \mathrm{NDVI_{min}} < NDVI < \mathrm{NDVI_{max}} \\ 1, & NDVI \geq \mathrm{NDVI_{max}} \end{cases} \tag{3}$$

where $\mathrm{NDVI_{max}}$ and $\mathrm{NDVI_{min}}$ are the thresholds for separating into vegetation areas, mixed areas, and bare soil areas. In order

to obtain globally consistent $f_v$ values, $\mathrm{NDVI_{max}}$ and $\mathrm{NDVI_{min}}$ were set to 0.5 and 0.2 (Sobrino et al., 2001), respectively.

According to Eqs. (2) and (3), the ASTER thermal channel emissivity of bare soil can be calculated as,

$$\varepsilon_{j,s}^{\mathrm{AST}} = \frac{\varepsilon_j^{\mathrm{AST}} - \varepsilon_{j,v}^{\mathrm{AST}} f_v}{1 - f_v} \tag{4}$$

where $\varepsilon_j^{\mathrm{AST}}, \varepsilon_{\lambda,v}^{\mathrm{AST}}$ and $\varepsilon_{\lambda,s}^{\mathrm{AST}}$ are the ASTER emissivity for the observation, dense vegetation, and bare soil in channel $j$

($j$=10,…,14), respectively. The ASTER thermal channel emissivities for dense vegetation is given in Meng et al. (2016).

In order to convert bare soil emissivities from ASTER spectral channels to AVHRR spectral channels, the following

linear relationship is fitted to channel emissivities obtained from the JHU bare soil spectral library (Salisbury, 1991):

$$\varepsilon_{i,s}^{AVH} = a_0 + a_1 \varepsilon_{10,s}^{\mathrm{AST}} + a_2 \varepsilon_{11,s}^{\mathrm{AST}} + a_3 \varepsilon_{12,s}^{\mathrm{AST}} + a_4 \varepsilon_{13,s}^{\mathrm{AST}} + a_5 \varepsilon_{14,s}^{\mathrm{AST}} \tag{5}$$

where $\varepsilon_{i,s}^{AVH}$ ($i$=4, 5) is the AVHRR bare soil emissivity in channel $i$ and $a_k$ ($k$=0,…,5) are coefficients (Table 6).

Figure 3 illustrates LSE estimation, which consists of two main parts:

Part I describes how static bare soil emissivity is obtained. After preparing the ASTER GED datasets, global mean NDVI

and channel emissivity maps are obtained and mean $f_v$ is calculated via Eq. (3). In combination with the LUT for ASTER

vegetation emissivity from Meng et al. (2016), an initial global ASTER bare soil emissivity map is obtained via Eq. (4).

However, due to regions with persistent cloud cover and over areas with dense vegetation (i.e. no visible bare soil fraction),





the obtained global emissivity maps for bare soil still have considerable data gaps. These missing values in the bare soil emissivity maps are filled with the average emissivity of the same soil type within the $3\times3$ neighbourhood pixels. If there is no neighbour valid pixel for averaging, the neighbourhood is enlarged until all data gaps are filled.

Part II describes the estimation of the daily dynamic emissivity. Firstly, the global ASTER background bare soil spectral channel emissivities are converted to AVHRR spectral channels via Eq. (5). Then, AVHRR channel emissivities are obtained via Eq. (2) with NDVI values from the AVH13C1 dataset. Vegetation emissivities are taken from a look-up table (Table 7), which is based on AVHRR LCTs and vegetation emissivities from Pinheiro et al. (2006). Furthermore, emissivities of built-up areas and water are used for separating these areas from other non-vegetated areas.

**3.4 Orbital Drift Correction**

The orbital drift of the NOAA-series satellites is a serious limitation for applications of AVHRR LST. Therefore, an orbital drift correction (ODC) would be highly useful and beneficial for many users. The overpass times of the NOAA-series afternoon satellites are between 13:00-17:30. In order to include the four-afternoon satellites (Table 1), the target time for ODC is set to 14:30 (solar time). According to the ODC method proposed by Liu et al. (2019a), the LST relationship between overpass time and ODC target time (14:30) can be written as:

$$T_{\mathrm{s}}(t) = T_{\mathrm{s}}(14.5) + T_{\mathrm{a}}\left\{\cos\left(\frac{\pi}{\omega}(t - t_{\mathrm{m}})\right) - \cos\left(\frac{\pi}{\omega}(14.5 - t_{\mathrm{m}})\right)\right\} \tag{6}$$

where $t$ is the time of the day in hours; $T_{\mathrm{a}}$ is the diurnal amplitude of LST in K; $\omega$ is the length of daytime and $t_{\mathrm{m}}$ is the time of maximum LST in hours (Göttsche and Olesen, 2001). Here, $\omega$ is determined by the duration of daytime: $\omega = \frac{2}{15}\cos^{-1}\left(\frac{\cos 85°}{\cos\phi\cos\delta} - \tan\phi\tan\delta\right)$, where $\phi$ is the latitude of the pixel in degree and $\delta$ is the solar declination. $\delta$ can be expressed as a function of the day of the year (DOY): $\delta = 23.45\sin\left(\frac{360°}{365}(284 + \mathrm{DOY})\right)$ (Elagib et al., 1998).

Similar to the component emissivity in Eq. (2), LST can be approximated as the weighted sum of the component temperatures of the vegetation and bare soil areas (Quan et al., 2018):

$$T \approx f_{\mathrm{v}}T_{\mathrm{veg}} + (1 - f_{\mathrm{v}})T_{\mathrm{soil}} \tag{7}$$

where $T_{\mathrm{veg}}$ and $T_{\mathrm{soil}}$ are the component temperatures of vegetation and bare soil, respectively.





Starting with the approach by Liu et al. (2019a), we further divide the diurnal temperature amplitude $T_a$ into two

components (i.e., vegetation and soil). Thus, Eq. (6) can then be rewritten as:

$$T_s(t) = f_v T_{s,veg}(14.5) + (1 - f_v)T_{s,soil}(14.5) + \left(f_v T_{a,veg} + (1 - f_v)T_{a,soil}\right)\left\{\cos\left(\frac{\pi}{\omega}(t - t_m)\right) - \cos\left(\frac{\pi}{\omega}(14.5 - t_m)\right)\right\} \quad (8)$$

where $T_{s,veg}(14.5)$, $T_{s,soil}(14.5)$ are the component temperatures of vegetation and bare soil at target time 14:30, respectively;

$T_{a,veg}$, $T_{a,soil}$ are the component of diurnal temperature amplitude $T_a$, respectively.

In Eq. (8), the parameters $T_s$, $f_v$, and $t$ are available for each pixel. To obtain the other five parameters $T_{s,veg}(14.5)$,

$T_{s,soil}(14.5)$, $T_{a,veg}$, $T_{a,soil}$, and $t_m$, it is assumed that the component temperatures and the shape of the diurnal temperature cycle

are approximately the same in a 3×3 pixel neighbourhood. With this assumption, there are nine equations to solve for the five

unknown parameters. To constrain the solution, boundaries are set for each parameter. The boundaries for $T_{s,veg}(14.5)$ and

$T_{s,soil}(14.5)$ are [$T_{center}$-10, $T_{center}$+15] K, and $T_{centre}$ is the LST for the centre pixel in the 3×3 neighbourhood. The boundaries for

$T_{a,veg}$ and $T_{a,soil}$ are [5, 40] K and $T_{a,soil}$ must be larger than $T_{a,veg}$. The boundary for $t_m$ is [12, 15] in hours. In order to obtain

more stable parameters, the pixel's ODC parameters obtained with an averaged value from the neighbourhood when Eq. (8)

cannot be fitted, e.g. $f_v$ are similar to each other (e.g. $f_v$=0, 1) in the 3×3 pixel neighbourhood. If there is no neighbour valid

pixel for averaging, the neighbourhood area is enlarged from 3×3 to 9×9. Once the parameters are determined, the LST at

14:30 can be calculated via Eq. (9).

$$T_s(14.5) = T_s(t) + \left(f_v T_{a,veg} + (1 - f_v)T_{a,soil}\right)\left\{\cos\left(\frac{\pi}{\omega}(14.5 - t_m)\right) - \cos\left(\frac{\pi}{\omega}(t - t_m)\right)\right\} \quad (9)$$

### 3.5 LST product validation based on *in-situ* LST

At the surface, *in-situ* LST can be estimated from measurements of broadband hemispherical upwelling radiance ($L_u$) and

atmospheric downwelling radiance ($L_d$) as

$$T_s = \sqrt[4]{\frac{L_u - (1-\varepsilon)L_d}{\varepsilon\sigma}} \quad (10)$$

where $T_s$ is *in-situ* LST; $\varepsilon$ is the broadband emissivity, which is calculated from AVHRR LSE for channel 4 and 5 via

$\varepsilon$=0.2489+0.2386$\varepsilon_4$+0.4998$\varepsilon_5$ (Liang, 2005); $\sigma$ (=5.67×$10^{-8}$ W/($m^2K^4$)) is the Stefan-Boltzmann constant.

Before the validation was performed, *in-situ* LST and AVHRR LST were accurately matched up in terms of geolocation



and acquisition time (nearest-neighbour interpolation and, depending on the site, time differences of less than 3 min, 15 min or 30 min). Furthermore, VZAs were limited to less than 40 °. Additionally, three-sigma ($3\sigma$) filtering (Eq. 11) was employed

to remove the samples contaminated by undetected clouds (Göttsche et al., 2016; Pearson, 2002).

$$S = 1.4826 * median\{|x_k - x^{med}|\} \tag{11}$$

where $x_k$ are the LST differences between the retrieved and *in-situ* values; $x^{med}$ is the median of the residuals. Matchups with residuals greater than $x^{med} +3S$ or less than $x^{med} - 3S$ are regarded as outliners.

## 4. Results and discussion

### 4.1 Training results and selection of SWAs

For NOAA-07/11 AVHRR, the candidate SWAs in Table 5 were already trained and evaluated by Zhou et al. (2019b). Here, the SWAs are additionally trained and evaluated for NOAA-09/14 AVHRR. Generally, the SWA training results for the four sensors are consistent with each other. The candidate algorithms OV1992, FO1996, and FOW 1996 show the worst regression accuracy regardless of atmospheric conditions with standard errors of the estimate (SEE) higher than 1.49 K, 1.48 K, and 1.32

K, respectively. For Warm-ATM, the SEE of PP1991 increases rapidly with increasing CWVC, while it shows good accuracy for Cold-ATM with SEE between 0.33 K and 0.75 K. The SEE values for UC1985 and MT2002 were larger than those of most other SWAs, even though they are still lower than for OV1992, FO1996, FOW1996, and PP1991: therefore, these six SWAs were disregarded in the further analysis. For the remaining 11 SWAs, a sensitivity analysis was performed for the TRA-G simulation dataset with uncertainties levels L1 and L2. The results showed that SO1991 and CO1994 are sensitive to

uncertainties in LSE and CWVC. Consequently, these two SWAs were also excluded from the candidate algorithm list. More details on the training and sensitivity analysis are provided in Zhou et al. (2019b).

The nine remaining SWAs for NOAA-09/14 AVHRR were then tested with the simulation datasets VAL-S and VAL-T. For completeness, Fig. 4 shows these results together with those obtained for NOAA-07/11. It can be seen that the retained nine SWAs have low RMSE values, which range between 0.38 K and 0.49 K for VAL-S and between 0.47 K and 0.68 K for

VAL-T. Since the atmospheric profiles used to generate VAL-S and VAL-T are globally distributed, we conclude that these



nine SWAs should perform well globally. The results for VAL-S in Fig. 4 reveal that BL-WD and WA2014 show the highest

overall accuracy, followed by BL1995, PR1984 and VI1991. The RMSE values of these four SWAs are ~0.48 K. For VAL-T,

BL1995 and BL_WD show the highest accuracy, followed by WA2014, VI1991, and PR1984; in this case, the RMSE of these

four SWAs is ~0.60 K. For all nine SWAs, the accuracy decreases as the VZA increases. While BL-WD achieves the highest

accuracy, no obvious differences between the other eight SWAs are observed. From the 17 LCTs over which the atmospheric

profiles are located, BL1995 performs best for six LCTs and BL-WD for three LCTs. In contrast, there is no LC type over

which VI1991 and UL1994 perform best. When assessing the effect of different atmospheric conditions, in Cold-ATM the

highest accuracy is found for BL1995 and BL-WD. In Warm-ATM, when $T_s$-NSAT is within [-4, 20] K, BL-WD performs the

best for CWVC below 2.5 g/cm$^2$, while PR1984 performs the best when CWVC exceeds 4.5 g/cm$^2$. When $T_s$-NSAT is within

[-16,4] K, WA2014 shows the best performance for CWVC below 3.5 g/cm$^2$; with increasing CWVC, BL1995 and BL-WD

show the highest accuracy. Overall, it is found that no single SWA achieves the highest accuracy under all conditions.

**4.2 Multi-LSTs ensemble**

The nine SWAs were integrated with the RF ensemble method. For comparison, the simple averaging (SA) method and

Bayesian Model Averaging (BMA) method were also employed. In contrast to Zhou et al. (2019b), here we used LST retrieved

with SWAs trained with TRA-G (L0, L1, and L2) and VAL-S/T (L0) to simulate a more realistic situation with uncertainty.

Generally, the MBE of the RF ensemble method and the BMA model method is negligible (of the order of 10$^{-4}$ K or less),

while the MBE of SA method and single SWA is larger (of the order of 0.1 K). It can be concluded that the two ensemble

methods (i.e. RF and BMA) similarly reduce systematic error. In terms of training accuracy, the RF model shows obvious

advantages with a STD of less than 0.50 K for the four NOAA AVHRR sensors while the STD of SA and BMA is larger and

varies between 1.27 and 1.35 K. Figure 5 highlights the importance of variance for forming the RF ensemble: the most

important SWA is BL1995 with an importance value of 0.67, 0.64, 0.68, and 0.83 for NOAA-07, 09, 11, and 14, respectively.

The second most important SWA for NOAA-07, 09, and 11 is ULW1994, while WA2014 is the second most important SWA

for NOAA-14. SR2000 is also of some importance for the ensemble process. Figure 5 confirms that the most important SWA,

i.e. BL1995, is consistent with the most accurate SWA under different atmospheric conditions.





Figure 6 shows the SEEs of the three ensemble methods for different CWVC zones and VZA subranges for NOAA-14

AVHRR. Compared to the BMA and SA models for all atmospheric conditions and VZAs, the RF ensemble model achieves

an obvious improvement in LST accuracy. For Cold-ATM, the SEE of RF increases slowly with increasing CWVC and VZA

and varies from 0.21 K to 0.45 K. In contrast, the SEEs of BMA and SA show larger variations for both, increasing CWVC

and VZA, and range from 0.72 K to 1.23 K. For Warm-ATM and CWVC less than 3.0 g/cm$^2$, there is no obvious increase in

SEE with increasing CWVC or VZA. However, SEE increases noticeably with increasing VZA when CWVC exceeds 3.0

g/cm$^2$, especially at VZA larger than 35 °. However, the SEE of RF is always smaller than that of BMA and SA: RF SEE only

exceeds 1.0 K when CWVC is larger than 5.0 g/cm$^2$ and VZA exceeds 60 °. Under the same conditions, the SEE of BMA and

SA is larger than 2.0 K. Therefore, it is concluded that the RF ensemble method achieves a higher training accuracy than the

BMA and SA methods, with a RF training accuracy of less than 1.0 K under most conditions.

To assess the stability and sensitivity of the RF model, the LST estimated with RF, BMA, and SA method were evaluated

against the VAL-T and VAL-S datasets at uncertainty levels L1 and L2. Figure 7 shows the evaluation of the three methods for

NOAA-14 AVHRR. For VAL-S at L1, STD and RMSE of about 0.7 K are found for all three methods; however, the biases of

RF (MBE=-0.04 K) and BMA (MBE=-0.03 K) are smaller than that of SA (MBE=-0.11 K) and negligible (i.e. less than ±0.1

K). For VAL-S at L2, the bias for all three methods is negligible. However, considerable improvements are obtained with the

RF method in terms of STD/RMSE, which is about 0.25K lower than for BMA and SA. For VAL-T at L1, the RF method has

a slightly larger bias (MBE=-0.1 K) than the SA/BMA methods; however, its STD/RMSE is smaller. For VAL-T at L2, the

three methods have a similar bias. However, RF has a significantly smaller STD/RMSE of 1.02/1.03 K than SA (1.41/1.42 K)

and BMA (1.38/1.39 K). Similar results were found for NOAA-07/09/11 AVHRR.

**4.3 Validation of RF-SWA LST against *in-situ* LST**

First, the generated RF-SWA LST was validated against *in-situ* LST from SURFRAD sites. Figure 8 shows a scatterplot

between RF-SWA LST and SURFRAD *in-situ* LST and some statistic indicators, i.e. MBE, RMSE, STD, $R^2$, and N (i.e. sample

size). High correlations are found between RF-SWA LST and *in-situ* LST with a $R^2$ range of 0.91-0.96. MBE varies between

-1.59 K and 2.71 K and RMSE between 2.25 K and 3.86 K. Compared to LST products for MODIS, AATSR, and VIIRS,





which were also validated against SURFRAD *in-situ* LST (Duan et al., 2019; Liu et al., 2019b; Martin et al., 2019), RF-SWA

LST shows a similar accuracy and precision. It should be noted that the large MBE at BND, GWN, and TBL are probably due

to a lack of *in-situ* LST representativeness at the satellite scale, e.g. BND, a seasonal bias variation is observed. During the

dormancy season, the surface within the ground radiometer's FOV and the corresponding AVHRR pixel are both fairly

homogeneously covered by bare soil and grassland, which leads to smaller LST differences. In contrast, during the growing

season, most of the area within the AVHRR pixel is covered by cropland and the fraction of vegetation cover depends on the

crop's growth stage: especially in the early growing and harvesting season, there are many bare areas between crop rows,

which causes larger LST differences. If only BND matchups during the dormancy season are considered, the corresponding

MBE and RMSE between RF-SWA LST and *in-situ* LST reduce to 1.56 K and 2.58 K, respectively. At GWN, the land cover

within the ground radiometer's FOV is also grassland; however, the corresponding AVHRR pixel includes several nearby

forest areas. Therefore, the daytime LST observed on the pixel scale tends to lower than the *in-situ* LST. At TBL, RF-SWA

LST is lower than *in-situ* LST for *in-situ* LST larger than 300 K: this may be explained by a larger vegetation area southeast

of the site, which is included in the AVHRR pixel, while the ground radiometer's entire FOV is covered by bare soil.

Figure 9 shows scatterplots between RF-SWA LST and NDBC lake surface water temperature (LSWT) and some statistic

indicators for buoys in the East Pacific, the Big Lakes, Gulf of Mexico, and western Atlantic. As shown in Table 4, the number

of buoys for each area differs. The RF-SWA LST shows a good correlation with *in-situ* LSWT with $R^2$ ranging from 0.94 to

0.98. The plots also show low systematic errors and high precision, e.g. MBEs are less than 0.26 K and RMSE ranges from

0.77 K to 0.89 K. Overall, the validation results resemble those obtained for the simulation datasets in section 4.2. Furthermore,

the validation results meet WMO's requirements for applications of LST/LSWT in different fields (WMO, 2020).

**4.4 Orbital Drift Correction**

The retrieved RF-SWA LST were normalized for the orbital drift of the NOAA-series satellites using the orbital drift correction

method described in section 3.4. Since water surface temperature varies relatively slowly, only the retrieved surface

temperatures over land were normalized. The orbital drift corrected LST (ODC LST) was then validated against the same *in-

situ* data as in section 4.3. Figure 10 shows a boxplot of the residuals ($T_{\text{AVHRR}}-T_{\text{in-situ}}$) for the ODC LST and RF-SWA LST. The



plot shows that the bias of the ODC LST over the 6 sites is similar to that of the uncorrected RF-SWA LST. From the six

SURFRAD sites, BND has the highest positive bias, while GWN and TBL show negative biases. Following the explanation

in section 4.3, this is probably due to less representative *in-situ* measurements. The standard deviations (STD) of the ODC

LST residuals at the six SURFRAD sites are 3.62 K (BND), 2.34 K (DRA), 3.38 K (FPK), 3.45 K (GWN), 2.57 K (PSU), and

3.69 K (TBL). The STD variations (ODC LST – RF-SWA LST) ranges from 0.06 K to 1.15 K. This indicates that the ODC

LST maintains the good accuracy of RF-SWA LST and its performance primarily depends on surface conditions. This is

understandable because the improved ODC method uses adjacent pixels to compensate for the lack of temporal information.

Nevertheless, the improved ODC method provides a practical way to correct the effect of orbital drift on LST retrieved from

NOAA AVHRR data.

**4.5 Global ODC LST product examples**

Figure 11 shows monthly averaged ODC LST for March, June, September, and December 1999 normalized to 14:30 solar time.

The LST show an obvious annual variation as seasons change with Earth's revolution around the Sun. In March and September

(Fig. 11 a and c), the Sun is overhead near the equator, which then receives most of the solar energy. However, the highest LST

are observed north and south of the equator, i.e. over the Sahara and Australia. This is due to the equatorial regions' dense

coverage with tropical rainforests, e.g. in the Amazon and Congo basins. The lowest LST are observed in the northern

hemisphere around 45 °N and around the Qinghai-Tibet Plateau. In June (Fig. 11 b), the Sun is more overhead in the northern

hemisphere and the area with the highest LST is located around 30 °N, e.g. over the Sahara Desert, the Arabian Peninsula, and

the Iran-Pamir Plateau. At the same time, LST also increases north of 45 °N. In December (Fig. 11 d), the area with the highest

LST is mainly located over Oceania and parts of South America. It should be noted that the large areas with invalid data, which

are mainly observed at latitudes larger than 45 °, are caused by the strict cloud filtering algorithms, which frequently recognize

snow and ice as cloud, and polar night when no visible data are available to calculate NDVI (related to LSE). Furthermore, in

June there are many invalid pixels over southern and southwestern China, which is regularly affected by cloudy weather.

In order to demonstrate the temporal consistency between satellites, Figure 12 shows time series of monthly averaged

ODC LST from 1981-2000 for the Amazon basin, the Arctic pole, and the Tibetan plateau (areas are shown in Fig. 1): no

significant orbital drift or inconsistencies can be seen, indicating that the ODC method adequately normalized the retrieved

AVHRR LST. The larger annual variations over the North pole (Fig. 12 b) are related to the specific variation of solar radiation

in high latitude areas, i.e. polar day and polar night. For the Amazon basin (Fig. 12 a), the North Arctic pole (Fig. 12 b), and

the Tibetan plateau (Fig. 12 c), the linear regressions (blue lines) show different trends with rates of $0.048 \pm 0.024$ K/year ($p$-

value=0.046), $0.087 \pm 0.221$ K/year ($p$-value=0.695), and $0.081 \pm 0.103$ K/year ($p$-value=0.433), respectively. However, these

rates may be affected by averaging over large areas and by the frequently missing data due to clouds. Therefore, more in-depth

analyses, especially with *in-situ* observations and reanalysis data, are needed. (Liu et al., 2008; Rigor et al., 2000; Schneider

and Hook, 2010; Wu et al., 2013).

**5. Data availability**

Global LST products retrieved from NOAA/AVHRR data between 1981 to 2000 are freely available at

https://doi.org/10.5281/zenodo.3934354 for RF-SWA LST (Ma et al., 2020a); https://doi.org/10.5281/zenodo.3936627 for

ODC LST (Ma et al., 2020c); https://doi.org/10.5281/zenodo.3936641 for monthly averaged LST (Ma et al., 2020b). The

dataset is also available at the National Earth System Science Data Center, National Science & Technology Infrastructure of

China (http://www.geodata.cn/thematicView/GLASS.html) and the University of Maryland (http://glass.umd.edu/LST/).

**6. Conclusion and Outlook**

Three global LST products with a spatial resolution of 0.05 °×0.05 ° have been generated from historical NOAA-7/9/11/14

AVHRR data (1981-2000). These LST products are obtained in four steps: (1) training and evaluation of 17 AVHRR SWAs,

(2) integrating nine selected SWAs with the Random Forest method (RF-SWA), (3) correcting the effect of orbital drift by

normalising RF-SWA LST to 14:30 solar time, and (4) validating the retrieved LST against *in-situ* LST data.

The 17 trained candidate SWAs generally showed consistent results for the four sensors. The candidate algorithms

OV1992, FO1996, FOW1996, PP1991, UC1985, and MT2002 had larger SEE than the other SWAs while SO1991 and

CO1994 were sensitive to uncertainties in LSE and CWVC. Therefore, these SWAs were rejected. The nine remaining SWAs



were evaluated based on the simulation datasets VAL-S and VAL-T. The results show that the trained nine SWAs have RMSE

ranging between 0.38 K and 0.55 K for VAL-S and between 0.53 K and 0.69 K for VAL-T. Since the atmospheric profiles used

to simulate and evaluate were chosen to be globally representative, we conclude that these nine SWAs should perform well

globally.

The RF ensemble method was then been applied to the nine selected SWAs. Compared to individual SWAs, sample

averaging, and the BMA ensemble method, the RF ensemble method showed the best accuracy when evaluated against the

simulation datasets. The RF ensemble algorithm yielded an accuracy better than 0.8 K for a maximum LSE uncertainty of 0.02

and a maximum CWVC uncertainty of 1.0 g/cm$^{-2}$; the accuracy was still better than 1.10 K when the maximum LSE uncertainty

increased to 0.04. Based on these results, the algorithm theoretically satisfies the target accuracy requirement of WMO, i.e. an

accuracy better than 1.0 K at a spatial resolution of 5 km. Furthermore, it is concluded that the RF method outperforms the SA

and BMA methods and has the greatest potential for improving LST retrieval accuracy.

The RF-SWA LST and ODC LST are validated against *in-situ* LST from SURFRAD sites and NDBC buoys. Against

SURFRAD LST, the MBE of RF-SWA LST varies from -1.59 K to 2.71 K and its STD varies from 2.26 K to 2.76 K, which

is similar to LST products retrieved from other sensors, e.g. MODIS. Against NDBC data from 1981-2000, RF-SWA LST also

shows good accuracy and precision: with its small MBE (less than 0.10 K) and a STD ranging from 0.84 to 1.05 K, its

performance against *in-situ* water temperature is similar to that for the simulated datasets. When validated against the same

SURFRAD LST, the MBE of ODC LST ranges from -1.05 K to 3.01 K, which is similar to the MBE obtained for RF-SWA

LST; its STD increases and ranges from 2.34 K to 3.69 K. Overall, it is concluded that both RF-SWA LST and ODC LST

achieve similar accuracy.

The generated global AVHRR LST is well suited to meet the needs of many applications and studies, e.g. global climate

change, radiation budget, and energy balance, mapping of land cover change. However, further research should address the

following points: first, the developed LST products were validated against *in-situ* LST data from North America, while they

need to be validated globally, e.g. against AsiaFlux measurements, historical air temperature, reanalysis data, etc. Second,

ODC LST is obtained at a single overpass time, which required using prior knowledge on temporal parameters. If additional

information on LST would be available, e.g. from modelling datasets, geostationary satellite datasets, and AVHRR nighttime

datasets. Third, over some areas, there are many invalid values, e.g. southwest China, which frequently experiences cloudy

and rainy weather. It is expected that future work can utilize recent progress in generating global all-weather LST products

(Martins et al., 2019; Zhang et al., 2019) to help integrating multi-source data, e.g. passive microwave brightness temperature

and reanalysis data.

**Author contributions**

Jin Ma and Ji Zhou designed the research, and Jin Ma implemented the research and completed the original manuscript; Ji

Zhou and Frank-Michael Göttsche supervised the research; Shunlin Liang founded and improved the research; Shaofei Wang

and Mingsong Li assisted in generating the RF-SWA LST and ODC LST, respectively. All co-authors revised the manuscript

and contributed to the writing.

**Competing interests**

The authors declare that they have no conflict of interest.

**Acknowledgements**

This work was supported by the National Natural Science Foundation of China (Grant 41871241) and by the Fundamental

Research Funds of the Central Universities of China (Grant ZYGX2019J069). Jin Ma also thanks China Scholarship Council

(Grant 201906070077) for his stay in Karlsruhe Institute of Technology, Germany. The NOAA AVHRR data were downloaded

from Level-1 and Atmosphere Archive & Distribution System Distributed Active Archive Center

(https://ladsweb.modaps.eosdis.nasa.gov/), SURFRAD data were downloaded from

ftp://aftp.cmdl.noaa.gov/data/radiation/surfrad/, BUOYS data were downloaded from National Data Buoy Center

(https://www.ndbc.noaa.gov/).



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




 **Tables**

**Table 1 Details of the selected AVHRR datasets**

| Name | Satellite | Start Date | End Date | Spatial resolution | Temporal resolution |
|------|-----------|------------|----------|--------------------|---------------------|
| | NOAA-07 | 1981/06/24 | 1985/02/02 | | |
| AVH02C1.465 | NOAA-09 | 1985/01/04 | 1988/11/07 | 0.05 °×0.05 ° | Daily daytime |
| AVH13C1.465 | NOAA-11 | 1988/11/07 | 1994/12/31 | | |
| | NOAA-14 | 1995/01/01 | 2000/10/31 | | |

**Table 2 Global LSE ranges determined from JHU spectral emissivity library for AVHRR channels 4 and 5**

| Satellite and sensor | LSE at 11 μm (channel 4) | LSE at 12 μm (channel 5) |
|----------------------|--------------------------|--------------------------|
| NOAA-07 AVHRR | 0.674 – 0.996 | 0.692 – 0.991 |
| NOAA-09 AVHRR | 0.665 – 0.996 | 0.713 – 0.991 |
| NOAA-11 AVHRR | 0.670 – 0.996 | 0.697 – 0.991 |
| NOAA-14 AVHRR | 0.672 – 0.994 | 0.661 – 0.991 |

 **Table 3 Atmospheric profile datasets and corresponding simulation datasets**

| Sources | Name | CWVC (g/cm²) | NSAT (K) | Number of profiles | VZA | $T_s$ | LSE | Sample size | Name of dataset |
|---------|------|--------------|----------|--------------------|-----|-------|-----|-------------|-----------------|
| SeeBor V5.0 | GAPD | 0.014 – 7.939 | 224.25 – 309.05 | 549 | 15 | 10 | 48 | 3952800 | TRA-G |
| | ATP-S | 0.005 – 4.999 | 201.96 – 313.50 | 4761 | 10 | 1 | 1 | 47610 | VAL-S |
| TIGR2000 V1.2 | ATP-T | 0.058 – 8.199 | 233.85 – 314.16 | 506 | 10 | 1 | 1 | 5060 | VAL-T |




**Table 4 SURFRAD sites and NDBC buoys used for LST validation**

| ID | Site | Network | Elevation | Latitude | Longitude | Sensor | LC type | Valid period |
|----|------|---------|-----------|----------|-----------|--------|---------|--------------|
| BND | Bondville, Illinois | SURFRAD | 230 | 40.0519 | -88.3731 | Eppley PIR | Cropland | 1995.01-2000.10 |
| DRA | Desert Rock, Nevada | SURFRAD | 1007 | 36.6237 | -116.0195 | Eppley PIR | Open Shrubland | 1998.03-2000.10 |
| FPK | Fort Peck, Montana | SURFRAD | 634 | 48.3078 | -105.1017 | Eppley PIR | Grassland | 1995.01-2000.09 |
| GWN | Goodwin Creek, Missisippi | SURFRAD | 98 | 34.2547 | -89.8729 | Eppley PIR | Wooded Grassland | 1995.01-2000.10 |
| PSU | Penn. State Univ., Pennsylvania | SURFRAD | 376 | 40.7201 | -77.9309 | Eppley PIR | Deciduous Broadleaf Forest | 1998.07-2000.10 |
| TBL | Table Mountain, Boulder, Colorado | SURFRAD | 1689 | 40.1250 | -105.2368 | Eppley PIR | Cropland | 1995.08-2000.08 |
| BEP | 46025, 46027, 46053, 46054 | NDBC | 0 | 33.763 – 60.587 | -119.053 – 146.833 | thermistors | Water | 1982.04-2000.10 |
| BGL | 45001, 45002, 45003, 45004, 45005, 45006, 45007, 45008 | NDBC | 175 | 41.677 – 48.061 | -82.398 – -89.793 | thermistors | Water | 1981.07-2000.10 |
| BGM | 42007, 42020, 42035 | NDBC | 0 | 26.968 – 30.09 | -88.32 – -96.693 | thermistors | Water | 1990.05-2000.10 |
| BWA | 41008, 41009, 44007, 44013, 44025 | NDBC | 0 | 28.508 – 43.525 | -70.141 – -80.868 | thermistors | Water | 1982.02-2000.10 |




**Table 5 Initial candidate Split Window Algorithms (SWAs)**

| Name | Formula | Reference |
|---|---|---|
| OV1992 | $T_s = A_0 + A_1 T_{11} + A_2(T_{11} - T_{12})$ | Ottlé and Vidal-Madjar (1992) |
| FO1996 | $T_s = A_0 + A_1 T_{11} + A_2(T_{11} - T_{12}) + A_3(T_{11} - T_{12})^2$ | Francois and Ottle (1996) |
| PR1984 | $T_s = A_0 + A_1 T_{11} + A_2(T_{11} - T_{12}) + A_3 T_{11}\varepsilon_{11} + A_4(T_{11} - T_{12})(1 - \varepsilon_{11})$ $+ A_5 T_{12}\Delta\varepsilon$ | Price (1984) |
| UC1985 | $T_s = A_0 + A_1 T_{11} + A_2(T_{11} - T_{12}) + A_3(1 - \varepsilon)$ | Ulivieri and Cannizzaro (1985) |
| BL-WD | $T_s = A_0 + \left(A_1 + A_2\dfrac{1-\varepsilon}{\varepsilon} + A_3\dfrac{\Delta\varepsilon}{\varepsilon^2}\right)(T_{11} + T_{12})$ $+ \left(A_4 + A_5\dfrac{1-\varepsilon}{\varepsilon} + A_6\dfrac{\Delta\varepsilon}{\varepsilon^2}\right)(T_{11} - T_{12})$ | Becker and Li (1990a) Wan and Dozier (1996) |
| PP1991 | $T_s = A_0 + A_1\dfrac{T_{11} - T_0}{\varepsilon_{11}} + A_2\dfrac{T_{12} - T_0}{\varepsilon_{12}} + A_3\dfrac{1-\varepsilon_{11}}{\varepsilon_{11}} + T_0$ | Prata and Platt (1991) |
| VI1991 | $T_s = A_0 + A_1 T_{11} + A_2(T_{11} - T_{12}) + A_3\dfrac{1-\varepsilon}{\varepsilon} + A_4\dfrac{\Delta\varepsilon}{\varepsilon}$ | Vidal (1991) |
| UL1994 | $T_s = A_0 + A_1 T_{11} + A_2(T_{11} - T_{12}) + A_3(1 - \varepsilon) + A_4\Delta\varepsilon$ | Ulivieri et al. (1994) |
| WA2014 | $T_s = A_0 + \left(A_1 + A_2\dfrac{1-\varepsilon}{\varepsilon} + A_3\dfrac{\Delta\varepsilon}{\varepsilon^2}\right)(T_{11} + T_{12})$ $+ \left(A_4 + A_5\dfrac{1-\varepsilon}{\varepsilon} + A_6\dfrac{\Delta\varepsilon}{\varepsilon^2}\right)(T_{11} - T_{12}) + A_7(T_{11} - T_{12})^2$ | Wan (2014) |
| FOW1996 | $T_s = A_0 + (A_1 w + A_2 w^2 + A_3)T_{11} + (A_4 w + A_5 w^2 + A_6)T_{12} + A_7 w + A_8 w^2$ | Francois and Ottle (1996) |
| SO1991 | $T_s = A_0 + A_1 T_{11} + [A_2 w + A_3 + (A_4 w + A_5)(1 - \varepsilon_{11}) + (A_6 w + A_7)\Delta\varepsilon](T_{11}$ $- T_{12}) + \dfrac{1-\varepsilon_{11}}{\varepsilon_{11}} T_{11}[A_8 w + A_9(A_{10} w + A_{11})\Delta\varepsilon]$ $- \dfrac{1-\varepsilon_{12}}{\varepsilon_{12}} T_{12}[A_{12} w + A_{13}(A_{14} w + A_{15})\Delta\varepsilon]$ | Sobrino et al. (1991) |
| ULW1994 | $T_s = A_0 + A_1 T_{11} + (A_2 w + A_3)(T_{11} - T_{12}) + (A_4 w + A_5)(1 - \varepsilon)$ $+ (A_6 w + A_7)\Delta\varepsilon$ | Ulivieri et al. (1994) |
| CO1994 | $T_s = A_0 + A_1 T_{11} + A_2(T_{11} - T_{12}) + A_3(T_{11} - T_{12})^2$ $+ [(A_4 w + A_5)T_{11} + (A_6 w + A_7)](1 - \varepsilon)$ $- [(A_8 w + A_9)T_{11} + (A_{10} w + A_{11})]\Delta\varepsilon$ | Coll et al. (1994) |
| SR2000 | $T_s = A_0 + A_1 T_{11} + A_2(T_{11} - T_{12}) + A_3(T_{11} - T_{12})^2 + (A_4 w + A_5)(1 - \varepsilon)$ $- (A_6 w + A_7)\Delta\varepsilon$ | Sobrino and Raissouni (2000) |
| MT2002 | $T_s = A_0 + A_1 T_{11} + A_2(T_{11} - T_{12}) + A_3(T_{11} - T_{12})^2 + (A_4 w + A_5)(1 - \varepsilon)$ | Ma and Tsukamoto (2002) |





| BL1995 | $T_s = A_0 + A_1 w + [A_2 + (A_3 w cos\theta + A_4)(1 - \varepsilon_{11}) - (A_5 w + A_6)\Delta\varepsilon](T_{11} + T_{12})$ $+ [A_7 + A_8 w + (A_9 + A_{10}w)(1 - \varepsilon_{11})$ $- (A_{11}w + A_{12})\Delta\varepsilon](T_{11} - T_{12})$ | Becker and Li (1995) |
|---|---|---|
| GA2008 | $T_s = A_0 + A_1 T_{11} + A_2(T_{11} - T_{12}) + A_3(T_{11} - T_{12})^2$ $+ (A_4 + A_5 w + A_6 w^2)(1 - \varepsilon) + (A_7 + A_8 w)\Delta\varepsilon$ | Galve et al. (2008) |

Note: subscripts 11 and 12 denote channels at approximately 11μm and 12μm, respectively, while $T_{11}$ and $T_{12}$ and $\varepsilon_{11}$ and $\varepsilon_{12}$ are their associated BTs and LSEs; $\varepsilon = (\varepsilon_{11} + \varepsilon_{12})/2$, $\Delta\varepsilon = (\varepsilon_{11} - \varepsilon_{12})$; $A_i$ are coefficients; $T_0$ in PP1991 is 273.15 K; $w$ is CWVC and $\theta$ is VZA.






**Table 6 Coefficients for converting bare soil emissivity from ASTER to AVHRR (see section 3.3).**

| Sensor | Channel | $a_0$ | $a_1$ | $a_2$ | $a_3$ | $a_4$ | $a_5$ | RMSE | $R^2$ |
|--------|---------|-------|-------|-------|-------|-------|-------|------|-------|
| NOAA-07 | 4 | 0.0000 | 0.0049 | -0.0071 | 0.0006 | 0.7749 | 0.2267 | 0.0001 | 0.99 |
| | 5 | 0.3064 | -0.1484 | 0.2676 | -0.0657 | -0.7622 | 1.3984 | 0.0016 | 0.91 |
| NOAA-09 | 4 | 0.0005 | 0.0041 | -0.0085 | 0.0029 | 0.8228 | 0.1781 | 0.0001 | 0.99 |
| | 5 | 0.2513 | -0.1392 | 0.2572 | -0.0757 | -0.7070 | 1.4102 | 0.0014 | 0.94 |
| NOAA-11 | 4 | 0.0007 | 0.0053 | -0.0091 | 0.0020 | 0.7895 | 0.2115 | 0.0001 | 0.99 |
| | 5 | 0.2944 | -0.1473 | 0.2666 | -0.0699 | -0.7404 | 1.3929 | 0.0016 | 0.92 |
| NOAA-14 | 4 | 0.0013 | -0.0083 | 0.0068 | 0.0042 | 0.8045 | 0.1912 | 0.0001 | 0.99 |
| | 5 | 0.3945 | -0.1591 | 0.2756 | -0.0467 | -0.8340 | 1.3647 | 0.0021 | 0.84 |

**Table 7 Emissivities of different vegetation types, water, and built-up surfaces for AVHRR channel 4 and channel 5.**

| LCTs | NO. | NOAA-07 AVHRR | | NOAA-09 AVHRR | | NOAA-11 AVHRR | | NOAA-14 AVHRR | |
|------|-----|-----------|-----------|-----------|-----------|-----------|-----------|-----------|-----------|
| | | channel 4 | channel 5 | channel 4 | channel 5 | channel 4 | channel 5 | channel 4 | channel 5 |
| Evergreen forest | 1, 2 | 0.989 | 0.988 | 0.990 | 0.987 | 0.989 | 0.988 | 0.990 | 0.987 |
| Deciduous forest | 3, 4 | 0.974 | 0.971 | 0.975 | 0.970 | 0.974 | 0.971 | 0.975 | 0.970 |
| Mixed forest | 5 | 0.982 | 0.979 | 0.983 | 0.979 | 0.982 | 0.979 | 0.983 | 0.979 |
| Woodland | 6 | 0.982 | 0.979 | 0.983 | 0.979 | 0.982 | 0.979 | 0.983 | 0.979 |
| Wooded grassland | 7 | 0.982 | 0.979 | 0.983 | 0.979 | 0.982 | 0.979 | 0.983 | 0.979 |
| Closed shrubland | 8 | 0.982 | 0.979 | 0.983 | 0.979 | 0.982 | 0.979 | 0.983 | 0.979 |
| Open shrubland | 9 | 0.982 | 0.979 | 0.983 | 0.979 | 0.982 | 0.979 | 0.983 | 0.979 |
| Grassland | 10 | 0.982 | 0.986 | 0.983 | 0.985 | 0.982 | 0.986 | 0.983 | 0.985 |
| Cropland | 11 | 0.982 | 0.986 | 0.983 | 0.985 | 0.982 | 0.986 | 0.983 | 0.985 |
| Water | 0 | 0.991 | 0.987 | 0.991 | 0.987 | 0.991 | 0.987 | 0.991 | 0.987 |
| Built-up surface | 12 | 0.948 | 0.953 | 0.948 | 0.953 | 0.948 | 0.953 | 0.948 | 0.953 |


**Figures**

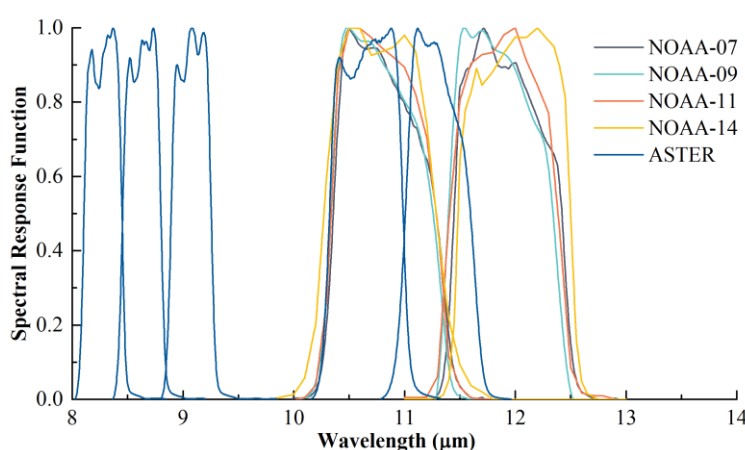

**Figure 1 Spectral response functions of NOAA-07/09/11/14 AVHRR and Terra ASTER.**

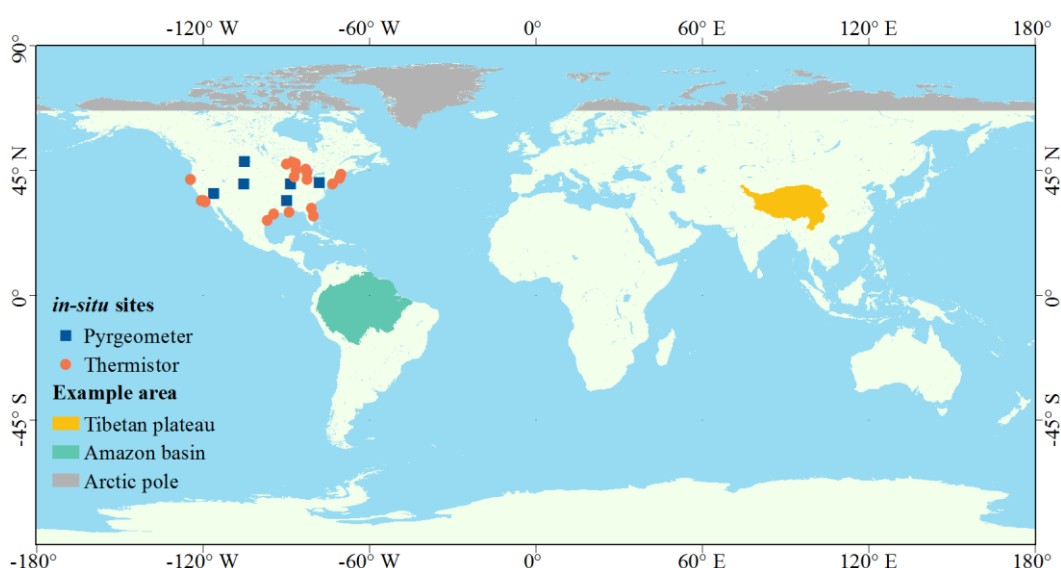

**Figure 2 Locations of SURFRAD sites and NDBC buoys, and the three sample areas. Blue squares indicate pyrgeometers; Red circles indicate contact thermistors.**





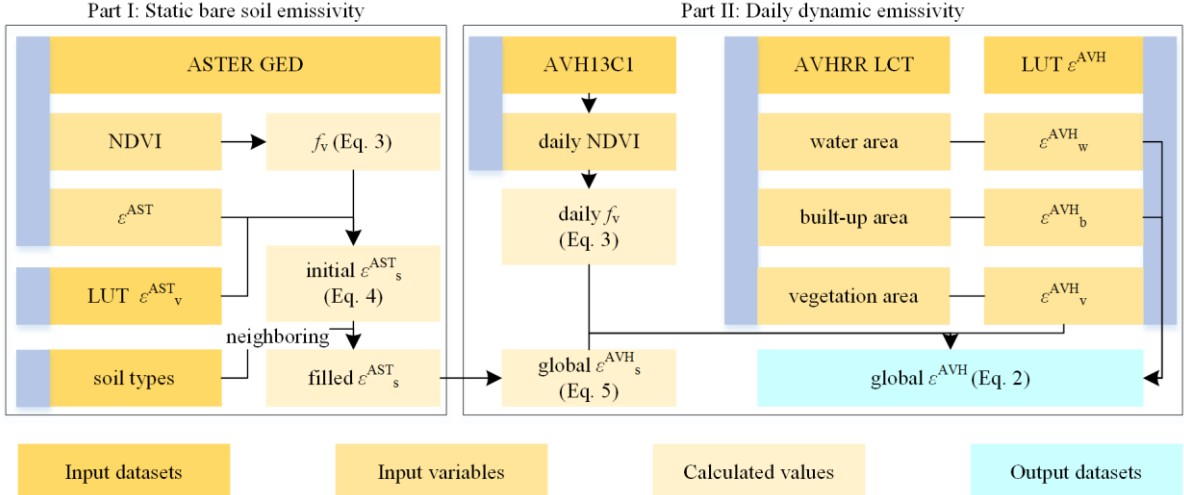

**Figure 3 Estimation of AVHRR LSE from ASTER GED, JHU spectral emissivity library data, LCT, and vegetation cover fraction.**

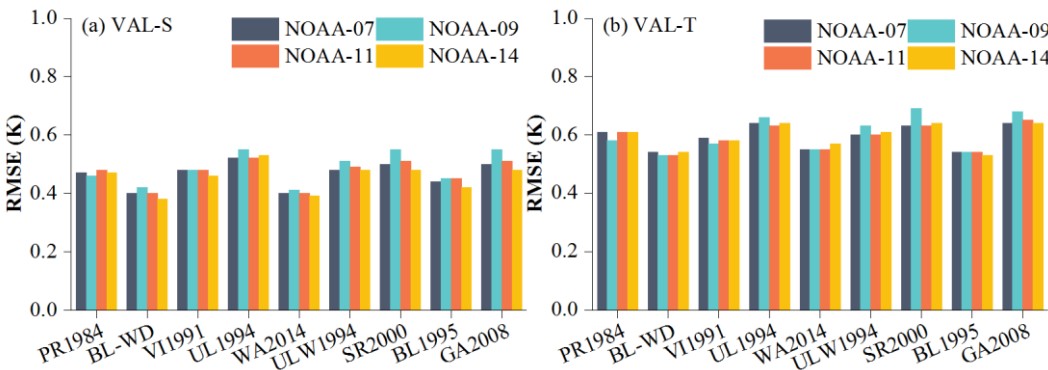

**Figure 4. Performance of the nine selected SWAs for simulation datasets VAL-S and VAL-T**

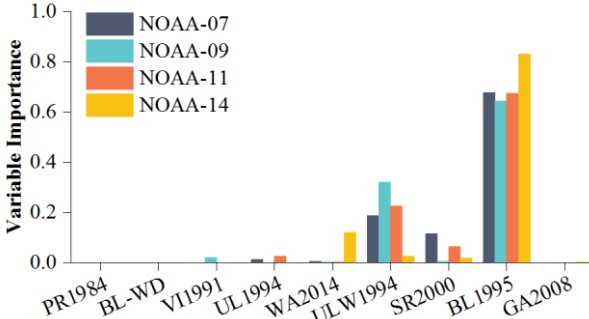

**Figure 5. Importance of the nine SWAs for the RF ensemble model**



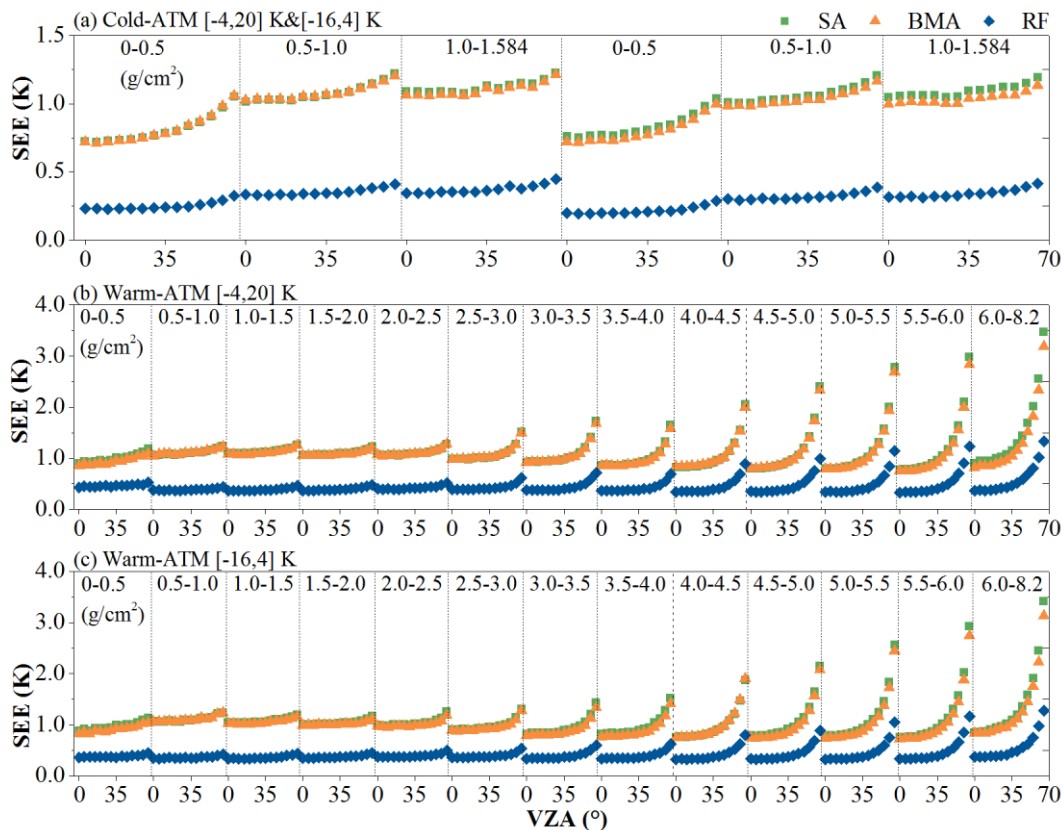

**Figure 6. SEE values of the three ensemble methods for NOAA-14 AVHRR under different atmospheric and VZA conditions. (a) Cold-ATM, for $T_s$–NSAT within [–4, 20] K (top) and [–16, 4] K (centre); (b) Warm-ATM, for $T_s$–NSAT within [–4, 20] K; (c) Warm-ATM, for $T_s$–NSAT within [–16, 4] K (bottom).**




**Figure 7. LST retrieved with the three ensemble methods for NOAA-14 against true LST. Results are based on simulation datasets VAL-T and VAL-S with added Gaussian noise (uncertainty levels L1 and L2).**





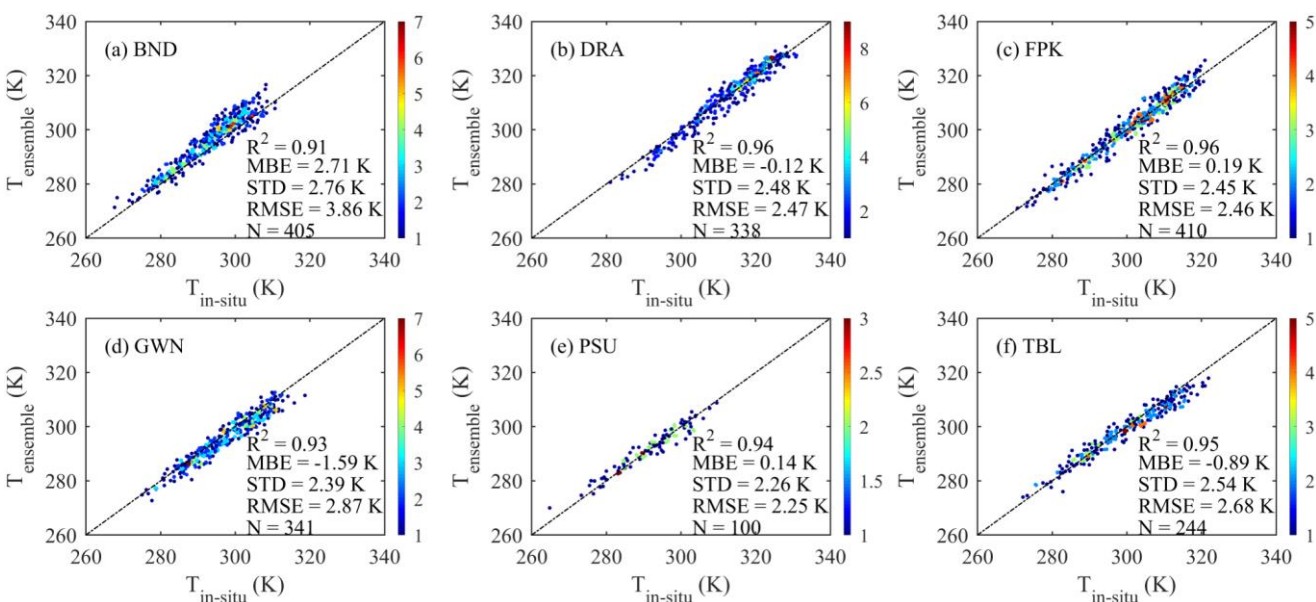

**Figure 8. RF-SWA LST against *in-situ* LST from SURFRAD sites.**

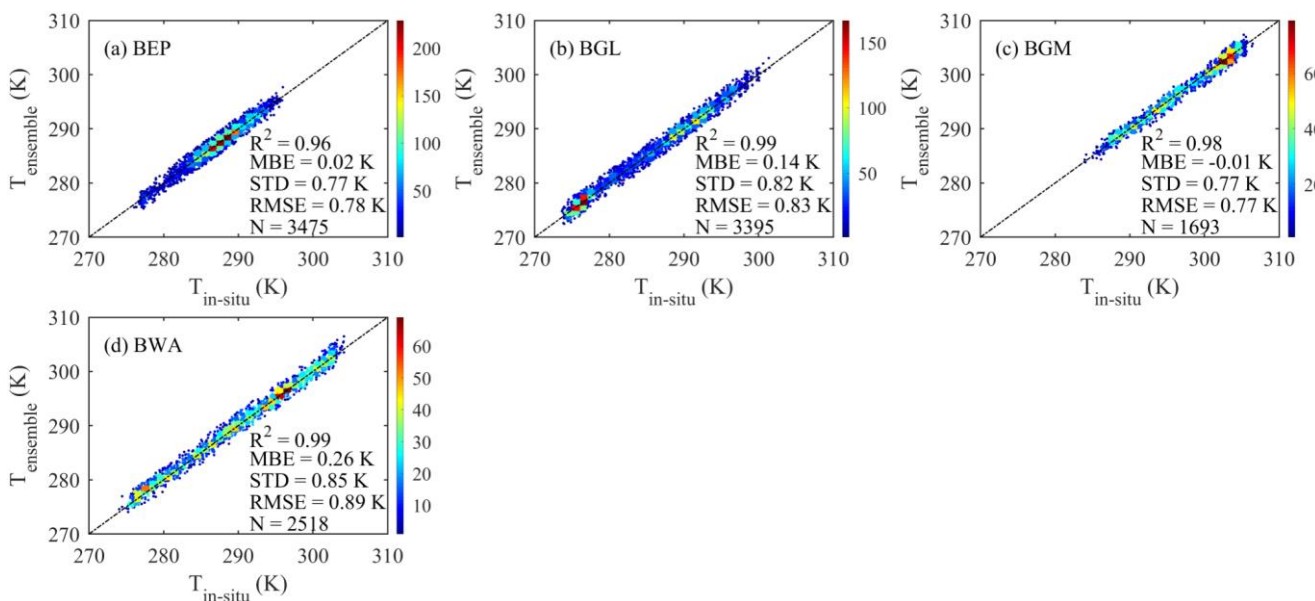

**Figure 9. RF-SWA LST against *in-situ* LSWT from four NDBC sites (buoy data) and corresponding statistics.**




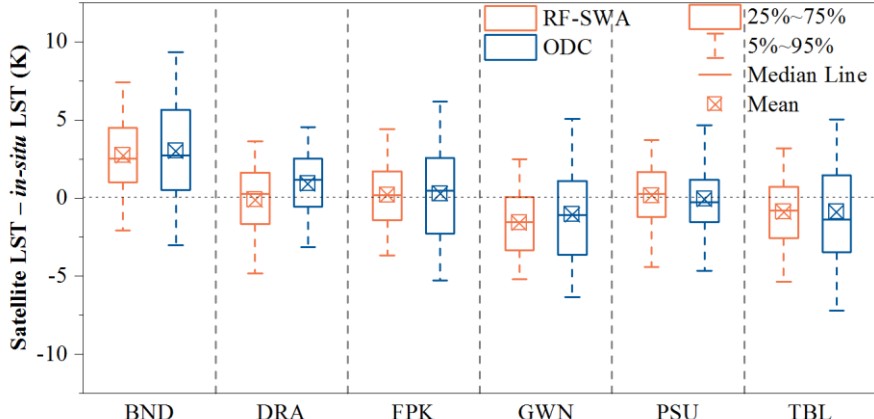

**Figure 10. Residuals w.r.t. *in-situ* LST for ODC and RF-SWA LST for six SURFRAD sites (BND, DRA, FPK, GWN, PSU, and TBL). Details on the sites are provided in Table 4.**

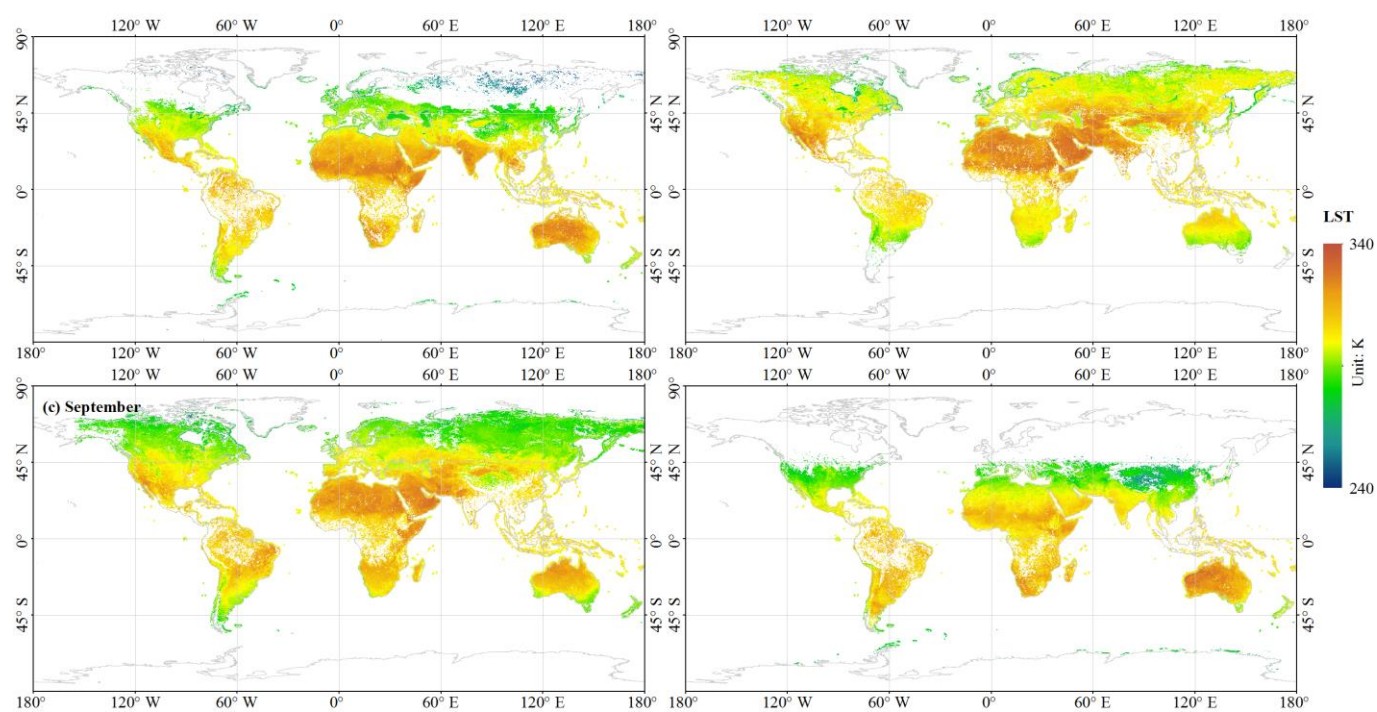


**Figure 11. Monthly averaged ODC LST retrieved from NOAA 14 data for 1999 normalized to 14:30: (a) March; (c) June; (c) September; (d) December.**





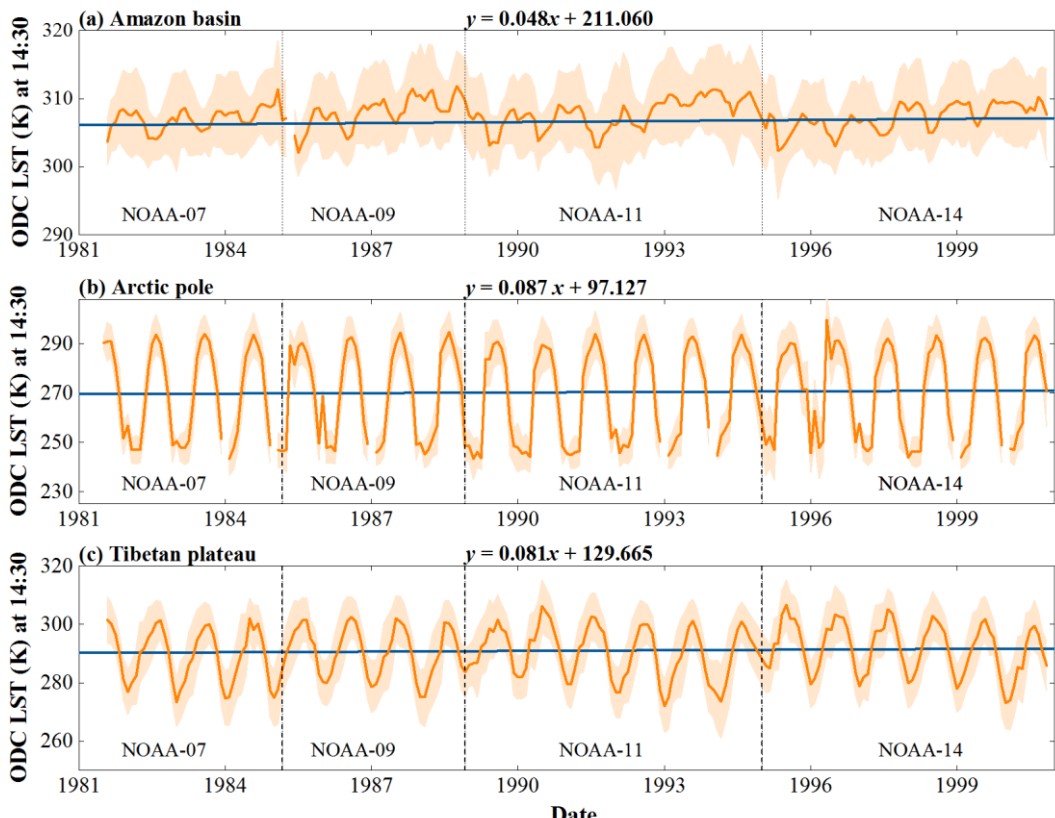

**Figure 12 Monthly averaged ODC LST time series normalized to 14:30 solar time for 1981-2000 over the Amazon basin (a), the North pole (b), and the Tibet plateau (c).**