# Peer review of "A global long-term (1981-2000) land surface temperature product for NOAA AVHRR"

_Earth System Science Data, 2020_

## Referee Comment (RC1) · Anonymous Referee #1 · 17 Aug 2020

This is an interesting manuscript described a processed AVHRR LST dataset that is very useful for long-term study of land surface temperature variation. Specifically, the dataset can be referred as a base for the climatological study of surface temperature since 2000: variation analyses between the LSTs derived from recent/current and the AVHRR LSTs can be a solid evidence of the global climate change. The method described in the manuscript can also be applied for producing long-term LST data record from other satellite missions, such as EOS MODIS and JPSS VIIRS.

The manuscript provides details of the LST algorithms being applied, multiple datasets being used, which are all good for readers to use the data, or to process their own long-term record of LST data.

Improvements suggested:

How the in-situ LST is estimated? Quality control/noise reduction in the process?

Cloud pixel exclusion (how cloud information was provided in the original data?) process?

The final 0.05 deg resolution data – Is this the resolution from the original AVHRR dataset? If not how about the compositing/aggregation process applied?

How the monthly average data set is generated? Details about the compositing/aggregation process?

End
* * *

---

## Short Comment (SC1) · 21 Aug 2020

Dear sir or madam,

thanks for the interesting paper. I have some question regarding your various uncertainty sources described on line 204f:

(1.) Why do you group them in two levels and what is the meaning/advantages of these levels? (2.) What is the explanation of $e_{11}$, $e_{12}$, and CWVC? Do they have any physical meaning? (3.) I assume you derived the thresholds for the uncertainty level in the paper Zhou et al. (2019b)?

Thank you and best regards, Andreas Baumann

---

## Referee Comment (RC2) · Anonymous Referee #2 · 3 Sep 2020

This manuscript reports a global land surface temperature dataset derived from the historical NOAA AVHRR data. According to my understanding, the most important contribution is that this global dataset was built over a long period of time from 1981 to 2000, and is needed by the scientific community in the field of geoscience. The second contribution is that the authors also conducted orbit-drift correction for the land surface temperature. With this dataset, I believe that the scientific community in the field of geoscience can better address the issues associated with climate change, hydrology, environment, etc. Therefore, this manuscript is definitely within the scope of ESSD. Additionally, this manuscript is well organized and written.

Nevertheless, I suggest that the authors consider the following comments and then improve the manuscript.

[Figure]

1. Line 38: the authors claim that the coarse resolution and high penetration depth are two main problems, which affect the accuracy of surface temperature from passive microwave remote sensing. However, the authors should keep in mind that the surface emissivity, as well as other physical mechanisms beyond our understanding, are also the main reasons. The authors need to clearly mention these points here.

2. Line 43-44: according to my experience, I also think that the algorithm selection depends on the availability of the required input parameters. I suggest revising here.

3. Line 46: There are many satellite sensors with both the 11 and 12 microns. I suggest mentioning the NOAA AVHRR and ENVISAT AATSR before SLSTR.

4. Line 57: The authors state 'no single SWA performs the best under all conditions'. How can you obtain such a conclusion? Please explain here and add supporting references here.

5. Line 58-59: The products cannot be retrieved. Also, cite the following references for the MODIS LST products: Wan, 2002, RSE; Wan, 2008, RSE.

6. Line 65: check the status of Sentinel-3C and make any necessary revision.

7. Line 82: I'm confused by 'cover progressively smaller areas'.

8. Line 97: I would suggest deleting the last sentence. It may appear in the wrong place. It should be in the methodology section instead of the introduction.

9. Line 117: Add references for the SST.

10. Line 139: How did you obtain these 48 land surface emissivities? Please explain. Such information is important for the authors.

11. Line 165: many studies use the SURFRAD data to validate the surface temperature derived from satellite data. I would suggest adding more references.

12. Line 180: you used the random forest to integrate multiple algorithms. Please

explain why you selected this method?

13. Line 185: how do you conduct the monthly averaging?

14. Line 192: please give the reason why you use 0.12 K?

15. Line 209-210: Please note that here is the method part, so you have not conducted the integration yet. Therefore, from the logical sequence, you don't know whether this method can get stable and robust results.

16. Line 243: as for the NDVI threshold method, I suggest citing Sobrino et al. (2008). Please check this reference.

17. Line 315: please simply give the results for NOAA-7 and NOAA-11 AVHRR here or somewhere.

18. Line 335: why the accuracy depends on the land cover type?

19. Line 397: what are the WMO requirements? Please explain.
* * *

---

## Author Comment (AC2) · 28 Sep 2020

Dear Dr. Andreas Baumann,

Thank you very much for your comments and questions.

Sorry for the missing explanation of $\epsilon_{11}$, $\epsilon_{12}$, and CWVC. In our context, they are the Land Surface Emissivitys (LSE) of NOAA AVHRR band centered at $11\mu$m and $12\mu$m, and column water vapour content, respectively. The study is a continuation of Zhou et al. (2019b) and, in order to be consistent with our previous study, the uncertainty thresholds were chosen to be similar to Zhou et al. (2019b).

LSEs and CWVC are the main input parameters of the Split-window Algorithm. Therefore, we analyzed the sensitivity of the candidate SWAs to these input parameters. We

used the NDVI-threshold method to determine the LSEs of a given pixel. A method comparison performed by Sobrino et al. (2001) yielded root-mean-square errors of 0.020 between the NDVI-threshold method and the Temperature-Independent Spectral Indices method and of 0.025 between the NDVI-threshold method the Thermal Infrared Radiance Ratio Model. Therefore, we increased the uncertainty of the NDVI-threshold method and the maximum uncertainty of LSE to 0.020 for level 1 (L1) and 0.040 for level 2 (L2). CWVC was derived from the Modern-Era Retrospective Analysis for Research and Applications (MERRA) dataset. Validation indicates that CWVC is biased by 0.24 g/cm$^2$ in the tropical zone (Rienecker et al., 2011). Considering the typical inhomogeneity within a MERRA grid, we increased the maximum CWVC uncertainty to 1.0 g/cm$^2$. This results in two combinations of maximum LSE uncertainty and CWVC uncertainty, which we grouped into two levels: (i) L1: $|\delta\epsilon_{11}|_{max}\leq$ 0.02, $|\delta\epsilon_{12}|_{max}\leq$ 0.02, and $|\delta CWVC|_{max}\leq$ 1.0 g/cm$^2$; (ii) L2: $|\delta\epsilon 11|_{max}\leq$ 0.04, $|\delta\epsilon 12|_{max}\leq$ 0.04, and $|\delta CWVC|_{max}\leq$ 1.0 g/cm$^2$.

**References:**

Sobrino, J. A., Raissouni, N. and Li, Z. L.: A comparative study of land surface emissivity retrieval from NOAA data, Remote Sens. Environ., 75(2), 256–266, doi:10.1016/S0034-4257(00)00171-1, 2001.

Rienecker, M. M., Suarez, M. J., Gelaro, R., Todling, R., Bacmeister, J., Liu, E., Bosilovich, M. G., Schubert, S. D., Takacs, L., Kim, G. K., Bloom, S., Chen, J., Collins, D., Conaty, A., Da Silva, A., Gu, W., Joiner, J., Koster, R. D., Lucchesi, R., Molod, A., Owens, T., Pawson, S., Pegion, P., Redder, C. R., Reichle, R., Robertson, F. R., Ruddick, A. G., Sienkiewicz, M. and Woollen, J.: MERRA: NASA's modern-era retrospective analysis for research and applications, J. Clim., 24(14), 3624–3648, doi:10.1175/JCLI-D-11-00015.1, 2011.

Thank you again and Best Regards,

Jin Ma and co-authors.

---

## Author Comment (AC3) · 28 Sep 2020

**A global long-term (1981-2000) land surface temperature product for NOAA AVHRR: Response to Referee 2**

Jin Ma, Ji Zhou, Frank-Michael Göttsche, Shunlin Liang, Shaofei Wang, Mingsong Li

*Correspondence to*: J. Zhou (jzhou233@uestc.edu.cn)

We would like to thank Referee 2 for her/his careful review of the manuscript and her/his constructive criticism and valuable comments. Comments by the referee are colored in black, our replies or comments are colored in blue.

This manuscript reports a global land surface temperature dataset derived from the historical NOAA AVHRR data. According to my understanding, the most important contribution is that this global dataset was built over a long period of time from 1981 to 2000, and is needed by the scientific community in the field of geoscience. The second contribution is that the authors also conducted orbit-drift correction for the land surface temperature. With this dataset, I believe that the scientific community in the field of geoscience can better address the issues associated with climate change, hydrology, environment, etc. Therefore, this manuscript is definitely within the scope of ESSD. Additionally, this manuscript is well organized and written.

Nevertheless, I suggest that the authors consider the following comments and then improve the manuscript.

Thank you for the positive evaluation and for stressing the usefulness of the data record.

1. Line 38: the authors claim that the coarse resolution and high penetration depth are two main problems, which affect the accuracy of surface temperature from passive microwave remote sensing. However, the authors should keep in mind that the surface emissivity, as well as other physical mechanisms beyond our understanding, are also the main reasons. The authors need to clearly mention these points here.

The uncertainty associated with land surface emissivity is indeed a problem in the LST retrieval from passive microwave data. We also agree that some physical mechanisms related to retrieving LST from passive microwaves are still beyond our understanding. However, uncertainty in thermal sampling depth is one of the largest sources of uncertainty in LST retrieval from passive microwaves. In fact, the physical definitions of land surface temperature (i.e. TIR LST) and the so-called retrieved temperature from passive microwave differ (Zhou et al., 2017). In the revised manuscript, we now additional mention possible uncertainties due to emissivity by writing ": compared to TIR remote sensing, it is limited by factors such as coarser spatial resolution, higher thermal sampling depth, and higher uncertainty in emissivity, which results in a lower retrieval accuracy (Zhou et al., 2017).".

**Reference:**

Zhou, J., Zhang, X., Zhan, W., Göttsche, F.M., Liu, S., Olesen, F.S., Hu, W., Dai, F., 2017. A Thermal Sampling Depth Correction Method for Land Surface Temperature Estimation From Satellite Passive Microwave Observation Over Barren Land. IEEE Trans. Geosci. Remote Sens. 55, 4743–4756. https://doi.org/10.1109/TGRS.2017.2698828

2. Line 43-44: according to my experience, I also think that the algorithm selection depends on the availability of the required input parameters. I suggest revising here.

We agree with you. We now state in the revised manuscript that "Selecting a suitable algorithm for retrieving LST depends on the sensor's number of TIR channels and their spectral specifications, as well as the available auxiliary input data.".

3. Line 46: There are many satellite sensors with both the 11 and 12 microns. I suggest mentioning the NOAA AVHRR and ENVISAT AATSR before SLSTR.

Thanks for your suggestion. We have changed the text in the revised manuscript to "The SWA is a good choice for retrieving LST from sensors with two or more TIR channels centred at 11 μm and 12 μm, e.g. Terra/Aqua MODIS, NOAA AVHRR, ENVISAT AATSR, and Sentinel-3 SLSTR.".

4. Line 57: The authors state 'no single SWA performs the best under all conditions'. How can you obtain such a conclusion? Please explain here and add supporting references here.

Thanks for your suggestion. The conclusion is based on previous research, e.g. Yu et al. (2009), Zhou et al. (2019), Yang et al. (2020). The SWA is the simplification of the radiative transfer model, which always depends on the available input parameters. Yu et al. (2009) and Zhou et al. (2019) compared seventeen SWAs developed by the scientific community in recent decades. The results show that some    SWAs achieve lower training accuracies and some are more sensitive to the input parameters. In other words, an SWA's performance depends on the application conditions, e.g. atmosphere conditions. Therefore, we concluded that no single SWA performs the best under all conditions.
**Reference:**
Yang, J., Zhou, J., Göttsche, F.-M., Long, Z., Ma, J. and Luo, R.: Investigation and validation of algorithms for estimating land surface temperature from Sentinel-3 SLSTR data, Int J Appl Earth Obs Geoinf., 91(April), 102136, doi:10.1016/j.jag.2020.102136, 2020.
Yu, Y., Tarpley, D., Privette, J. L., Goldberg, M. D., Rama Varma Raja, M. K., Vinnikov, K. Y. and Hui Xu: Developing Algorithm for Operational GOES-R Land Surface Temperature Product, IEEE Trans. Geosci. Remote Sens., 47(3), 936–951, doi:10.1109/TGRS.2008.2006180, 2009.
Zhou, J., Liang, S., Cheng, J., Wang, Y. and Ma, J.: The GLASS Land Surface Temperature Product, IEEE J. Sel. Top. Appl. Earth Obs. Remote Sens., 12(2), 493–507, doi:10.1109/JSTARS.2018.2870130, 2019b.

5. Line 58-59: The products cannot be retrieved. Also, cite the following references for the MODIS LST products: Wan, 2002, RSE; Wan, 2008, RSE.

Thanks for your comments. We changed the expression and added the suggested references for MODIS LST and now write "Currently, several LST products derived from satellite TIR remote sensing are available. Global LST products for Terra/Aqua MODIS are available since 2000, e.g. MOD11/MYD11 (Wan, 2008, 2014; Wan et al., 2002) and MOD21/MYD21 (Hulley and Hook, 2011).".
**Reference:**
Hulley, G. C. and Hook, S. J.: Generating consistent land surface temperature and emissivity products between ASTER and MODIS data for earth science research, IEEE Trans. Geosci. Remote Sens., 49(4), 1304–1315, doi:10.1109/TGRS.2010.2063034, 2011.
Wan, Z.: New refinements and validation of the MODIS Land-Surface Temperature/Emissivity products, Remote Sens. Environ., 112(1), 59–74, doi:10.1016/j.rse.2006.06.026, 2008.

Wan, Z.: New refinements and validation of the collection-6 MODIS land-surface temperature/emissivity product, Remote Sens. Environ., 140, 36–45, doi:10.1016/j.rse.2013.08.027, 2014.

Wan, Z., Zhang, Y., Zhang, Q. and Li, Z-L: Validation of the land-surface temperature products retrieved from terra moderate resolution imaging spectroradiometer data, Remote Sens. Environ., 83(1–2), 163–180, doi:10.1016/S0034-4257(02)00093-7, 2002.

6. Line 65: check the status of Sentinel-3C and make any necessary revision.

Sentinel-3C is still unavailable, i.e. a revision is currently not required.

7. Line 82: I'm confused by 'cover progressively smaller areas'.

With this sentence, we wanted to express that most glaciers on the Tibetan Plateau are in retreat and the areas covered by them are getting smaller and smaller. We changed the sentence to "…, e.g. most glaciers on the Tibetan Plateau are in retreat and the areas covered by them are getting smaller and smaller (Yao et al., 2012)."

**Reference:**

Yao, T., Thompson, L., Yang, W., Yu, W., Gao, Y., Guo, X., Yang, X., Duan, K., Zhao, H., Xu, B., Pu, J., Lu, A., Xiang, Y., Kattel, D. B. and Joswiak, D.: Different glacier status with atmospheric circulations in Tibetan Plateau and surroundings, Nat. Clim. Chang., 2(9), 663–667, doi:10.1038/nclimate1580, 2012.

8. Line 97: I would suggest deleting the last sentence. It may appear in the wrong place. It should be in the methodology section instead of the introduction.

The respective sentence has been deleted.

9. Line 117: Add references for the SST.

Thanks for your suggestion: we added a reference for SST in the revised manuscript.

10. Line 139: How did you obtain these 48 land surface emissivities? Please explain. Such information is important for the authors.

In this study, channel-effective emissivity was obtained from the Johns Hopkins University (JHU) spectral emissivity library by convolving emissivity spectra with the spectral response functions of NOAA-07/09/11/14 AVHRR, please see lines 141-143.

11. Line 165: many studies use the SURFRAD data to validate the surface temperature derived from satellite data. I would suggest adding more references.

Thanks for your suggestion; we added more references in the revised manuscript.

12. Line 180: you used the random forest to integrate multiple algorithms. Please explain why you selected this method?

Thanks for your question. In line 180 we only mention the general method for integrating multiple algorithms, i.e. the random forest (RF) method, to inform the reader about the basic concepts used in this study. The RF method is then briefly discussed in section 3.2. It has several advantages, including the ability to process large databases with high efficiency, unbiased estimation, and especially minimizing the risk of overfitting in explaining complicated nonlinear relationships when compared with detailed analytic expressions (Hutengs and Vohland, 2016), and it has been used in many studies, e.g land cover

classification (Rodriguez-Galiano et al., 2012), land surface parameter downscaling (Zhao et al., 2018), and estimating vegetation cover parameters (Mutanga et al., 2012). Please see lines 211-216.

**Reference:**

Hutengs, C. and Vohland, M.: Downscaling land surface temperatures at regional scales with random forest regression, Remote Sens. Environ., 178, 127–141, doi:10.1016/j.rse.2016.03.006, 2016.

Mutanga, O., Adam, E. and Cho, M. A.: High density biomass estimation for wetland vegetation using worldview-2 imagery and random forest regression algorithm, Int. J. Appl. Earth Obs. Geoinf., 18(1), 399–406, doi:10.1016/j.jag.2012.03.012, 2012.

Rodriguez-Galiano, V. F., Ghimire, B., Rogan, J., Chica-Olmo, M. and Rigol-Sanchez, J. P.: An assessment of the effectiveness of a random forest classifier for land-cover classification, ISPRS J. Photogramm. Remote Sens., 67(1), 93–104, doi:10.1016/j.isprsjprs.2011.11.002, 2012.

Zhao, W., Sánchez, N., Lu, H. and Li, A.: A spatial downscaling approach for the SMAP passive surface soil moisture product using random forest regression, J. Hydrol., 563(June), 1009–1024, doi:10.1016/j.jhydrol.2018.06.081, 2018.

13. Line 185: how do you conduct the monthly averaging?

The monthly data set is simply averaged from the daily orbital drift corrected LST. In detail, the program first searches the date labels of the daily LST data files to identify the data within the month to be processed. Then the sum of all valid LST within this month is calculated and divided by the number of valid LST. This step is now also explained in the revised manuscript.

14. Line 192: please give the reason why you use 0.12 K?

The design goals for the AVHRR thermal infrared channels were a NEΔT of 0.12K (@ 300K). Therefore, we added a Gaussian-distributed noise with a NEΔT of 0.12 K to simulate BTs measured by satellites more realistically.

15. Line 209-210: Please note that here is the method part, so you have not conducted the integration yet. Therefore, from the logical sequence, you don't know whether this method can get stable and robust results.

Thanks for your comment. We agree with you. In the study, we want to develop a more stable method to generate global LST. We want to integrate multiple single SWAs to reduce the random error in LST retrieval, which is primarily due to uncertainty in the input parameters. The respective sentence in the revised manuscript has been changed accordingly.

16. Line 243: as for the NDVI threshold method, I suggest citing Sobrino et al. (2008).
Please check this reference.

Thanks for your suggestion. The NDVI threshold method in Sobrino et al. (2008) uses the square of normalized NDVI to calculate the fraction of vegetation. However, in this study, we only use the normalized NDVI instead of its square, which refers to method proposed in Carlson and Ripley (1997).

**Reference:**

Carlson, T. N. and Ripley, D. A.: On the relation between NDVI, fractional vegetation cover, and leaf area index, Remote Sens. Environ., 62(3), 241–252, doi:10.1016/S0034-4257(97)00104-1, 1997

Sobrino, J. A., Jiménez-Muñoz, J. C., Sòria, G., Romaguera, M., Guanter, L., Moreno, J., Plaza, A. and Martínez, P.: Land surface emissivity retrieval from different VNIR and TIR sensors, IEEE Trans. Geosci.

Remote Sens., 46(2), 316–327,doi:10.1109/TGRS.2007.904834, 2008

17. Line 315: please simply give the results for NOAA-7 and NOAA-11 AVHRR here or somewhere.

Thanks for your suggestion. The results for NOAA-7 and NOAA-11 AVHRR are very similar to those obtained for NOAA-9 and NOAA-14. Therefore, we did not provide the respective results for NOAA-7 and NOAA-11. However, in the original manuscript we stated that "Generally, the SWA training results for the four sensors are consistent with each other.". In the revised manuscript, we also added a sentence explaining that the results for NOAA-14 AVHRR are a good example for the other sensors.

18. Line 335: why the accuracy depends on the land cover type?

There are two main reasons: on the one hand, the SWAs perform differently over different land covers, i.e. they show different training accuracies over the same land cover, even for the same input data. On the other hand, the nine SWAs were selected because they are relatively insensitive to the main input parameters, i.e. LSE and CWVC; however, the nine SWAs still differ in their sensitivity to uncertainty in the input parameters. In the validation with the simulation data, the LSE was set according to the land cover type over which the atmospheric profile was located. Therefore, the accuracy of each SWA depends on land cover type. The findings from the simulation data motivated us to look for a more suitable optimization method, e.g. random forests, in order to reduce dependence on input parameter uncertainty.

19. Line 397: what are the WMO requirements? Please explain.

Thanks for your suggestion. The WMO provides requirements for LST in several related fields, e.g. the uncertainty requirement is 2.0 K for Agricultural Meteorology and 1.0 K for Climate Monitoring. In the revised manuscript we added the sentence "Furthermore, the validation results meet WMO's requirements for applications of LST/LSWT in different fields, e.g. an uncertainty of 2.0 K for Agricultural Meteorology and of 1.0 K for Climate Monitoring (WMO, 2020)."
**Reference:**
WMO: Requirements defined for Land surface temperature, [online] Available from: https://www.wmo-sat.info/oscar/variables/view/96 (Accessed 29 July 2020), 2020.

---

## Author Response (AR1)

**Response to Referee 1**

**Jin Ma, Ji Zhou, Frank-Michael Göttsche, Shunlin Liang, Shaofei Wang, Mingsong Li**

Correspondence to: J. Zhou (jzhou233@uestc.edu.cn)

We would like to thank Referee 1 for her/his careful review of the manuscript and her/his constructive criticism and valuable comments. Comments by the referee are colored in black, our replies or comments are colored in blue.

This is an interesting manuscript described a processed AVHRR LST dataset that is very useful for long-term study of land surface temperature variation. Specifically, the dataset can be referred as a base for the climatological study of surface temperature since 2000: variation analyses between the LSTs derived from recent/current and the AVHRR LSTs can be a solid evidence of the global climate change. The method described in the manuscript can also be applied for producing long-term

- LST data record from other satellite missions, such as EOS MODIS and JPSS VIIRS. The manuscript provides details of the LST algorithms being applied, multiple datasets being used, which are all good for readers to use the data, or to process their own long-term record of LST data.
- 15 Thank you for the positive evaluation and for stressing the usefulness of the data record.

Improvements suggested:

How the in-situ LST is estimated? Quality control/noise reduction in the process?

In this study, the *in-situ* LST were collected from SURFRAD sites and NDBC. The NDBC data consisted of water surface temperatures measured directly by buoys: since these are highly accurate and quality controlled by NDBC (https://www.ndbc.noaa.gov/qc.shtml), we used the water temperatures as they were distributed.

The SURFRAD *in-situ* LST were calculated from measured broadband hemispherical upwelling radiance  $(L_u)$  and atmospheric downwelling radiance  $(L_d)$  using Stefan-Boltzmann's law:

$$T_{\rm s} = \sqrt[4]{\frac{L_{\rm u} - (1 - \varepsilon)L_{\rm d}}{\varepsilon\sigma}}$$

25

10

where broadband emissivity  $\varepsilon$  is obtained from AVHRR LSE in channels 4 and 5 via the empirical relationship  $\varepsilon$ =0.2489+0.2386 $\varepsilon$ 4+0.4998 $\varepsilon$ 5 (Liang, 2005) and  $\sigma$  (=5.67×10-8 W/(m2K4)) is the Stefan-Boltzmann constant.

Quality control is an integral part of the design and operation of the SURFRAD network, which results in datasets of high quality and well-defined measurement uncertainties (https://www.esrl.noaa.gov/gmd/grad/surfrad/). SURFRAD data have been directly used in satellite retrieved LST validation (Liu et al., 2019; Martin et al., 2019; Wang and Liang, 2009); therefore, we did not perform additional quality control or noise reduction. However, the *in-situ* LST and the satellite retrieved LST may still contain outliers, e.g. samples contaminated by undetected clouds. Therefore, three-sigma ( $3\sigma$ ) filtering was employed to

30

remove such possible outliers from the match-ups(Göttsche et al., 2016; Pearson, 2002). Please see section 3.5 for the corresponding description for the *in-situ* LST estimation and validation, and section 2.4 for the description of quality control in the revised manuscript.

**35 **Reference:**

40

Göttsche, F.-M., Olesen, F.-S., Trigo, I., Bork-Unkelbach, A. and Martin, M.: Long Term Validation of Land Surface Temperature Retrieved from MSG/SEVIRI with Continuous in-Situ Measurements in Africa, Remote Sens., 8(5), 410, doi:10.3390/rs8050410, 2016.

Liang, S.: Estimation of Surface Radiation Budget: I. Broadband Albedo, in Quantitative Remote Sensing of Land Surfaces, pp. 310–344. John Wiley & Sons. Inc., Hoboken, NJ, USA., 2005.

Liu, X., Tang, B. H., Yan, G., Li, Z. L. and Liang, S.: Retrieval of global orbit drift corrected land surface temperature from long-term AVHRR data, Remote Sens., 11(23), 2843, doi:10.3390/rs11232843, 2019a. Martin, M., Ghent, D., Pires, A., Göttsche, F.-M., Cermak, J. and Remedios, J.: Comprehensive In Situ Validation of Five Satellite Land Surface Temperature Data Sets over Multiple Stations and Years, Remote Sens., 11(5), 479,

**45 doi:10.3390/rs11050479, 2019.**

Wang, K. and Liang, S.: Evaluation of ASTER and MODIS land surface temperature and emissivity products using long-term surface longwave radiation observations at SURFRAD sites, Remote Sens. Environ., 113(7), 1556-1565. doi:10.1016/j.rse.2009.03.009, 2009.

50

**Cloud pixel exclusion (how cloud information was provided in the original data?) process?**

Thanks for your comment. In this study, we used the LTDR AVHRR dataset as the source data to produce the LST products. The dataset provides quality control (QC) flags for each pixel and contains information on clouds as well as other conditions,

55 e.g. cloud shadow, water, etc. When generating the LST products, we used the QC flags to identify pixel containing cloud and cloud shadow and excluded them from the processing. Please see line 316-317 in the revised manuscript.

The final 0.05 deg resolution data – Is this the resolution from the original AVHRR dataset? If not how about the compositing/aggregation process applied?

60

65

In this study, the LTDR AVHRR dataset served as the source data of the LST product. The spatial resolution of the LTDR AVHRR dataset is  $0.05^{\circ} \times 0.05^{\circ}$  (already processed from AVHRR's native resolution; please see Table 1). In order to clarify this, we added the following sentence to the revised manuscript (line 123-125):

"In this study, the AVHRR datasets from Long-Term Datasets Records (Pedelty et al., 2007) (LTDR, https://ltdr.modaps.eosdis.nasa.gov/) are used, including AVH02C1 and AVH13C1, for which spatial resolution has been processed to  $0.05^{\circ} \times 0.05^{\circ}$  (Table 1)."

**Reference:**

Pedelty, J., Devadiga, S., Masuoka, E., Brown, M., Pinzon, J., Tucker, C., Vermote, E., Prince, S., Nagol, J., Justice, C., Roy, D., Ju, J., Schaaf, C., Liu, J., Privette, J. and Pinheiro, A.: Generating a long-term land data record from the AVHRR and

Pearson, R. K.: Outliers in process modeling and identification, IEEE Trans. Control Syst. Technol., 10(1), 55-63, doi:10.1109/87.974338, 2002.

70

75

80

**How the monthly average data set is generated? Details about the compositing/aggregation process?**

The monthly average data are obtained from daily orbital drift corrected LST as follows: the program first searches the date labels of the daily LST data files to identify the data within the month to be processed. Then the sum of all valid LST within this month is calculated and divided by the number of valid LST. We added some text explaining this and other LST processing steps in a new section (section 3.5) of the revised manuscript:

"3.5 Generation of LST products

The product generation executable (PGE) code includes four Modules. Module I is for generating the multi-LST with the selected SWAs. Three different types of input data enter this module: (i) the satellite data: BTs from AVH02C1, NDVI from AVH13C1, bare soil emissivity (see section 3.3), and AVHRR LCTs from UMD; (ii) look-up tables; coefficients of the SWAs (see section 3.1), emissivities of vegetation, water, and built-up areas (see Table 7); and (iii) ancillary data: NSAT and CWVC from MERRA and land-sea mask. The QC flags in AVHR02C1 are also used to identify cloudy pixel. If a pixel contains cloud or cloud shadow, its LST is not calculated. Therefore, the output of Module I is multi-LST under clear sky conditions.

Module II is for integrating the multi-LST with the trained RF ensemble model. The inputs include the multi-LST from Module I and the RF ensemble model; the output is the ensemble LST, which is termed RF-SWA LST. Module III is for 85 normalizing the LST affected by orbital drift to 14:30 solar time. In this Module, the input datasets include the RF-SWA LST and NDVI; the latter is used for calculating the fraction of vegetation. The output of Module III is orbital drift corrected LST, which is termed OCD LST. Module IV is for generating monthly average LST: the module first groups ODC LST by month, sums the valid LST in each month up, and divides them by the respective number of valid LST. The output from this Module is monthly averaged ODC LST. All LST data are stored in standard HDF-EOS format. Table 8 shows the variables provided in the three types of LST data files."

90

**Response to Referee 2**

**Jin Ma, Ji Zhou, Frank-Michael Göttsche, Shunlin Liang, Shaofei Wang, Mingsong Li**

Correspondence to: J. Zhou (jzhou233@uestc.edu.cn)

We would like to thank Referee 2 for her/his careful review of the manuscript and her/his constructive criticism and 5 valuable comments. Comments by the referee are colored in black, our replies or comments are colored in blue.

This manuscript reports a global land surface temperature dataset derived from the historical NOAA AVHRR data. According to my understanding, the most important contribution is that this global dataset was built over a long period of time from 1981 to 2000, and is needed by the scientific community in the field of geoscience. The second contribution is that the authors also conducted orbit-drift correction for the land surface temperature. With this dataset, I believe that the scientific community in the field of geoscience can better address the issues associated with climate change, hydrology, environment, etc. Therefore, this manuscript is definitely within the scope of ESSD. Additionally, this manuscript is well organized and written. Nevertheless, I suggest that the authors consider the following comments and then improve the manuscript.

**15 Thank you for the positive evaluation and for stressing the usefulness of the data record.**

1. Line 38: the authors claim that the coarse resolution and high penetration depth are two main problems, which affect the accuracy of surface temperature from passive microwave remote sensing. However, the authors should keep in mind that the surface emissivity, as well as other physical mechanisms beyond our understanding, are also the main reasons. The authors need to clearly mention these points here.

20

10

The uncertainty associated with land surface emissivity is indeed a problem in the LST retrieval from passive microwave data. We also agree that some physical mechanisms related to retrieving LST from passive microwaves are still beyond our understanding. However, uncertainty in thermal sampling depth is one of the largest sources of uncertainty in LST retrieval from passive microwaves. In fact, the physical definitions of land surface temperature (i.e. TIR LST) and the so-called retrieved

25 temperature from passive microwave differ (Zhou et al., 2017). In the revised manuscript (line 37-39), we now additional mention possible uncertainties due to emissivity by writing ": compared to TIR remote sensing, it is limited by factors such as coarser spatial resolution, higher thermal sampling depth, and higher uncertainty in emissivity, which results in a lower retrieval accuracy (Zhou et al., 2017).".

Thanks for your suggestion. The results for NOAA-7 and NOAA-11 AVHRR are very similar to those obtained for
 NOAA-9 and NOAA-14. Therefore, we did not provide the respective results for NOAA-7 and NOAA-11. However, in the original manuscript we stated that "Generally, the SWA training results for the four sensors are consistent with each other.". In the revised manuscript, we also added a sentence explaining that the results for NOAA-14 AVHRR are a good example for the other sensors, please see line 345.

**165 18. Line 335: why the accuracy depends on the land cover type?**

There are two main reasons: on the one hand, the SWAs perform differently over different land covers, i.e. they show different training accuracies over the same land cover, even for the same input data. On the other hand, the nine SWAs were selected because they are relatively insensitive to the main input parameters, i.e. LSE and CWVC; however, the nine SWAs still differ in their sensitivity to uncertainty in the input parameters. In the validation with the simulation data, the LSE was set

170 according to the land cover type over which the atmospheric profile was located. Therefore, the accuracy of each SWA depends on land cover type. The findings from the simulation data motivated us to look for a more suitable optimization method, e.g. random forests, in order to reduce dependence on input parameter uncertainty.

**19. Line 397: what are the WMO requirements? Please explain.**

175 Thanks for your suggestion. The WMO provides requirements for LST in several related fields, e.g. the uncertainty requirement is 2.0 K for Agricultural Meteorology and 1.0 K for Climate Monitoring. In the revised manuscript we added the sentence "Furthermore, the validation results meet WMO's requirements for applications of LST/LSWT in different fields, e.g. an uncertainty of 2.0 K for Agricultural Meteorology and of 1.0 K for Climate Monitoring (WMO, 2020)." Please see lines 427-428.

40

Thank you again and Best Regards, Jin Ma and co-authors.

[revised manuscript text omitted]
_1w + A_2w^2 + A_3)T_{11} + (A_4w + A_5w^2 + A_6)T_{12} + A_7w + A_8w^2$                                                                        | Francois and Ottle (1996)         |
| 801991  | $I_{s} = A_{0} + A_{1}I_{11} + [A_{2}W + A_{3} + (A_{4}W + A_{5})(1 - \varepsilon_{11}) + (A_{6}W + A_{7})\Delta\varepsilon](I_{11})$                          |                                   |
|         | $-T_{12}) + \frac{1 - \varepsilon_{11}}{\varepsilon_{11}} T_{11} [A_8 w + A_9 (A_{10} w + A_{11}) \Delta \varepsilon]$                                         | Sobrino et al. (1991)             |
|         | $-\frac{1-\varepsilon_{12}}{\varepsilon_{12}}T_{12}[A_{12}w+A_{13}(A_{14}w+A_{15})\Delta\varepsilon]$                                                          |                                   |
| ULW1994 | $T_s = A_0 + A_1 T_{11} + (A_2 w + A_3)(T_{11} - T_{12}) + (A_4 w + A_5)(1 - \varepsilon)$                                                                     | Ulivieri et al. (1994)            |
| 001004  | $+ (A_6 W + A_7) \Delta \varepsilon$                                                                                                                           |                                   |
| CO1994  | $I_s = A_0 + A_1 I_{11} + A_2 (I_{11} - I_{12}) + A_3 (I_{11} - I_{12})^2$                                                                                     | $C_{11} + 1_{(1004)}$             |
|         | + $[(A_4W + A_5)I_{11} + (A_6W + A_7)](1 - \varepsilon)$                                                                                                       | Coll et al. (1994)                |
| GD 2000 | $- [(A_{8}W + A_{9})I_{11} + (A_{10}W + A_{11})]\Delta \mathcal{E}$                                                                                            | Caluring and Delegant             |
| SK2000  | $I_{s} = A_{0} + A_{1}I_{11} + A_{2}(I_{11} - I_{12}) + A_{3}(I_{11} - I_{12})^{2} + (A_{4}w + A_{5})(1 - \varepsilon) - (A_{6}w + A_{7})\Delta\varepsilon$    | (2000)                            |
| MT2002  | $T_s = A_0 + A_1 T_{11} + A_2 (T_{11} - T_{12}) + A_3 (T_{11} - T_{12})^2 + (A_4 w + A_5)(1 - \varepsilon)$                                                    | Ma and Tsukamoto
(2002)        |
|         |                                                                                                                                                                |                                   |

BL1995
$$T_{s} = A_{0} + A_{1}w + [A_{2} + (A_{3}w\cos\theta + A_{4})(1 - \varepsilon_{11}) - (A_{5}w + A_{6})\Delta\varepsilon](T_{11} + T_{12}) \\ + [A_{7} + A_{8}w + (A_{9} + A_{10}w)(1 - \varepsilon_{11}) - (A_{11}w + A_{12})\Delta\varepsilon](T_{11} - T_{12})$$

GA2008
$$T_{s} = A_{0} + A_{1}T_{11} + A_{2}(T_{11} - T_{12}) + A_{3}(T_{11} - T_{12})^{2} + (A_{4} + A_{5}w + A_{6}w^{2})(1 - \varepsilon) + (A_{7} + A_{8}w)\Delta\varepsilon$$

Galve et al. (2008)

Note: subscripts 11 and 12 denote channels centred at approximately 11µm and 12µm, respectively, while  $T_{11}$  and  $T_{12}$  and  $\varepsilon_{11}$ and  $\varepsilon_{12}$  are their associated BTs and LSEs;  $\varepsilon = (\varepsilon_{11}+\varepsilon_{12})/2$ ,  $\Delta \varepsilon = (\varepsilon_{11}-\varepsilon_{12})$ ;  $A_i$  are coefficients;  $T_0$  in PP1991 is 273.15 K; w is CWVC and  $\theta$  is VZA.

795 Table 6 Coefficients for converting bare soil emissivity from ASTER to AVHRR (see section 3.3).

| Sensor   | Channel           | a 0 | a 1 | a 2 | a 3 | a 4 | a 5 | RMSE   | R 2 |
|----------|-------------------|------------|------------|-----------------------|------------|------------|------------|--------|-----------------------|
|          | centred at        |            |            |                       |            |            |            |        |                       |
|          | 11µm 4     | 0.0000     | 0.0049     | -0.0071               | 0.0006     | 0.7749     | 0.2267     | 0.0001 | 0.99                  |
| NOAA-07  | 12µm <del>5</del> | 0.3064     | -0.1484    | 0.2676                | -0.0657    | -0.7622    | 1.3984     | 0.0016 | 0.91                  |
|          | 11µm 4     | 0.0005     | 0.0041     | -0.0085               | 0.0029     | 0.8228     | 0.1781     | 0.0001 | 0.99                  |
| NOAA-09  | 12µm5             | 0.2513     | -0.1392    | 0.2572                | -0.0757    | -0.7070    | 1.4102     | 0.0014 | 0.94                  |
|          | 11µm 4     | 0.0007     | 0.0053     | -0.0091               | 0.0020     | 0.7895     | 0.2115     | 0.0001 | 0.99                  |
| NOAA-11  | 12µm5             | 0.2944     | -0.1473    | 0.2666                | -0.0699    | -0.7404    | 1.3929     | 0.0016 | 0.92                  |
|          | 11µm 4     | 0.0013     | -0.0083    | 0.0068                | 0.0042     | 0.8045     | 0.1912     | 0.0001 | 0.99                  |
| INUAA-14 | 12µm5             | 0.3945     | -0.1591    | 0.2756                | -0.0467    | -0.8340    | 1.3647     | 0.0021 | 0.84                  |

Table 7 Emissivities of different vegetation types, water, and built-up surfaces for AVHRR channel centred at 11µm and 12µm<del>channel 4 and channel 5</del>.

|                  |      | NOAA-0'             | 7 AVHRR             | NOAA-09             | ) AVHRR             | NOAA-11             | AVHRR               | NOAA-14             | 4 AVHRR             |
|------------------|------|---------------------|---------------------|---------------------|---------------------|---------------------|---------------------|---------------------|---------------------|
| LCTs             | NO.  | 11µm <del>eha</del> | 12µm <del>cha</del> | 11µm <del>cha</del> | 12µm <del>cha</del> | 11µm <del>eha</del> | 12µm <del>eha</del> | 11µm <del>cha</del> | 12µm <del>cha</del> |
|                  |      | nnel 4              | <del>nnel 5</del>   | nnel 4              | <del>nnel 5</del>   | nnel 4              | <del>nnel 5</del>   | nnel 4              | nnel 5              |
| Evergreen forest | 1, 2 | 0.989               | 0.988               | 0.990               | 0.987               | 0.989               | 0.988               | 0.990               | 0.987               |
| Deciduous forest | 3,4  | 0.974               | 0.971               | 0.975               | 0.970               | 0.974               | 0.971               | 0.975               | 0.970               |
| Mixed forest     | 5    | 0.982               | 0.979               | 0.983               | 0.979               | 0.982               | 0.979               | 0.983               | 0.979               |
| Woodland         | 6    | 0.982               | 0.979               | 0.983               | 0.979               | 0.982               | 0.979               | 0.983               | 0.979               |
| Wooded grassland | 7    | 0.982               | 0.979               | 0.983               | 0.979               | 0.982               | 0.979               | 0.983               | 0.979               |
| Closed shrubland | 8    | 0.982               | 0.979               | 0.983               | 0.979               | 0.982               | 0.979               | 0.983               | 0.979               |
| Open shrubland   | 9    | 0.982               | 0.979               | 0.983               | 0.979               | 0.982               | 0.979               | 0.983               | 0.979               |
| Grassland        | 10   | 0.982               | 0.986               | 0.983               | 0.985               | 0.982               | 0.986               | 0.983               | 0.985               |
| Cropland         | 11   | 0.982               | 0.986               | 0.983               | 0.985               | 0.982               | 0.986               | 0.983               | 0.985               |
| Water            | 0    | 0.991               | 0.987               | 0.991               | 0.987               | 0.991               | 0.987               | 0.991               | 0.987               |
| Built-up surface | 12   | 0.948               | 0.953               | 0.948               | 0.953               | 0.948               | 0.953               | 0.948               | 0.953               |

|--|

| LST Name                         | Variable   | Description                                                         | Unit   | Data   | Scale | Offset | Dimension |
|----------------------------------|------------|---------------------------------------------------------------------|--------|--------|-------|--------|-----------|
|                                  |            | 2000-000                                                            | 0      | type   |       | 011500 |           |
|                                  | LST        | Land surface temperature                                            | Κ      | Uint16 | 0.02  | -      | 3600*7200 |
| RF-SWA
(instantaneous)
LST | View_time  | Time of LST observation (UTC)                                       | hrs    | Uint8  | 0.1   | -      | 3600*7200 |
|                                  | View_angle | View zenith angle                                                   | degree | Uint8  | -     | -      | 3600*7200 |
|                                  | QA         | LST quality flag                                                    | -      | Uint8  | -     | -      | 3600*7200 |
|                                  | Latitude   | Latitude                                                            | degree | Uint8  | -     | -      | 3600*1    |
|                                  | Longitude  | Longitude                                                           | degree | Uint8  | -     | -      | 7200*1    |
| ODC LST                          | LST        | Land surface temperature at 14:30 solar time                        | K      | Uint16 | 0.02  | -      | 3600*7200 |
|                                  | Latitude   | Latitude                                                            | degree | Uint8  | -     | -      | 3600*1    |
|                                  | Longitude  | Longitude                                                           | degree | Uint8  | -     | -      | 7200*1    |
| monthly
averaged LST          | LST        | Land Surface Temperature
averaged monthly at 14:30
solar time | K      | Uint16 | 0.02  | -      | 3600*7200 |
|                                  | Count      | the number of available pixels in a month                           | -      | Uint8  | -     | -      | 3600*7200 |
|                                  | Latitude   | Latitude                                                            | degree | Uint8  | -     | -      | 3600*1    |
|                                  | Longitude  | Longitude                                                           | degree | Uint8  | -     | -      | 7200*1    |

**Figures**

805 Figure 1 Spectral response functions of NOAA-07/09/11/14 AVHRR and Terra ASTER.